# JanusPipe: Efficient Pipeline Parallel Training for Machine Learning Interatomic Potentials

Hongyu Wang [* 1 2]   Weijian Liu [* 1 3]   Hongtao Xu [1 2]   Yan Wang [1 3]   Mingzhen Li [1]   Weile Jia [1]   Guangming Tan [1 3]

## Abstract

Discovering atom-level phenomena requires molecular dynamics (MD) simulations with ab initio accuracy. Machine learning interatomic potentials (MLIPs) enable stable, high-accuracy MD simulations, and their models exhibit scaling-law trends similar to large language models. However, the lack of scalable and efficient distributed training systems for conservative MLIPs makes them difficult to scale. This is because conservative MLIPs inherently follow a double-backward execution pattern, which involves computing gradients during the forward pass. This pattern creates a mismatch with existing distributed training systems, especially for pipeline parallelism. Therefore, we present JanusPipe, an efficient 3D-parallel (PP/DP/GP) training system tailored for conservative MLIPs. It integrates SymFold to enable memory-efficient pipeline parallelism for conservative MLIPs, and WaveK to reduce pipeline bubbles by balancing the four-phase compute time. Experimental results on 32 GPUs show that JanusPipe improves throughput by $1.51\times$ and $1.45\times$ on average over 1F1B and Hanayo, respectively.

## 1. Introduction

Molecular dynamics (MD) with *ab initio* accuracy is central to scientific discovery in emerging domains, such as energy-efficient batteries (Zheng et al., 2024), chemical catalysts (Tran et al., 2023), and drug design (Qiao et al., 2025). However, *ab initio* molecular dynamics based on

solving the Kohn-Sham equations requires solving the eigenvalue problem, whose computational cost scales as $O(N^3)$ with the number of atoms $N$. This high computational cost confines simulations to picosecond timescales and system sizes of at most a few thousand atoms (Jia et al., 2013a;b). Recent advances in the AI for Science ecosystem enable the development of universal machine-learning interatomic potentials (MLIPs) with near-linear scaling (approximately $O(N)$), substantially accelerating MD simulations (Jia et al., 2020; Batzner et al., 2022). Recent studies suggest that scaling up MLIPs by increasing training data and the parameter count can improve accuracy and generalization (Wood et al., 2025; Zhang et al., 2025; Li et al., 2025a; Bigi et al., 2026), echoing the scaling-law trends observed in large language models (LLMs) (Kaplan et al., 2020).

Conservative MLIPs output potential energy $E$ and compute forces as the negative gradient of $E$ with respect to atomic positions $x$, i.e., $F = -\nabla_x E$, yielding a conservative force field (Batatia et al., 2022; Deng et al., 2023; Qu & Krishnapriyan, 2024; Fu et al., 2025; Mazitov et al., 2025). In contrast, non-conservative MLIPs predict forces directly via an individual readout head. In practice, conservative MLIPs are widely adopted in many state-of-the-art MLIP designs (Riebesell et al., 2023; Bigi et al., 2025), because they can reduce drift over long-timescale simulations. During each training iteration, conservative MLIPs exhibit a **double-backward** execution pattern (**second-order**) that differs from first-order training workloads (e.g., LLMs), as shown in Figure 1. This pattern contains four phases: Forward Energy (FE), Forward Force (FF), Backward Force (BF), and Backward Energy (BE). Note that FF is executed during the forward pass, although it computes $\partial E / \partial x$ via automatic differentiation. Hence, we still regard FF as part of the forward pass.

The double-backward execution pattern creates a mismatch with existing distributed training systems, particularly pipeline parallelism (PP) (Narayanan et al., 2021). Specifically, existing PP schedules target first-order workloads and assume a single forward pass followed by a single backward pass per micro-batch (Figure 1(a)). Naively adapting first-order PP schedules to second-order MLIPs leads to two performance issues. First, there are more data dependencies within the forward pass (i.e., FF needs the activations

---

[*]Equal contribution [1]SKLP, Institute of Computing Technology, Chinese Academy of Sciences, Beijing, China [2]School of Advanced Interdisciplinary Sciences, University of Chinese Academy of Sciences, Beijing, China [3]University of Chinese Academy of Sciences, Beijing, China. Correspondence to: Mingzhen Li <limingzhen@ict.ac.cn>, Weile Jia <jiaweile@ict.ac.cn>, Guangming Tan <tgm@ict.ac.cn>.

*Proceedings of the $43^{rd}$ International Conference on Machine Learning*, Seoul, South Korea. PMLR 306, 2026. Copyright 2026 by the author(s).

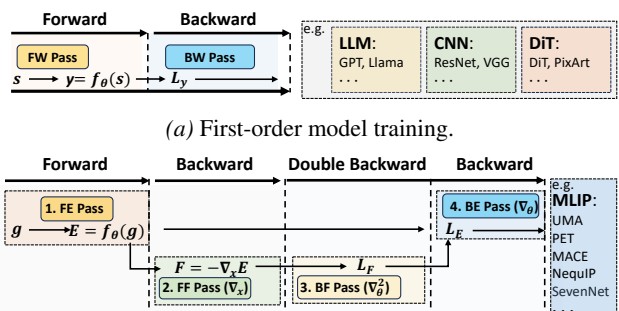

*(a)* First-order model training.

*(b)* Conservative MLIP training (second-order).

*Figure 1.* (a) First-order workloads perform one forward pass and one backward pass per micro-batch. (b) Conservative MLIPs compute forces by differentiating the predicted energy in the forward stage ($F = -\nabla_x E$), which introduces a double-backward execution pattern with four phases (FE/FF/BF/BE). See Table 1 for notation.

of FE), triggering redundant recomputation and parameter replication across PP stages. Second, the execution times of different phases are model-dependent, although their values satisfy a consistent partial order (e.g., $t_{FE} < t_{FF} < t_{BE} < t_{BF}$), thereby breaking the sophisticated overlap of PP schedules.

To tackle the above issues, we propose JanusPipe, a distributed training system tailored for conservative MLIPs. We can abstract the PP execution on each device into an instruction list. Building on the instruction list, JanusPipe introduces *SymFold*, a schedule transformation that converts a first-order pipeline schedule into a four-phase pipeline schedule for conservative MLIPs. Then JanusPipe reorders the instruction list into a *WaveK* schedule, which can reduce pipeline bubbles under a controllable memory footprint. In addition, we further combine JanusPipe with data parallelism (Li et al., 2020) and graph parallelism (Sriram et al., 2022). Experimental results on 32 GPUs show that JanusPipe improves end-to-end training throughput by $1.51\times$ over 1F1B (Narayanan et al., 2021) and $1.45\times$ over Hanayo (Liu et al., 2023) on average, and reduces peak GPU memory by up to $20.56\%$ and $42.70\%$ respectively.

Specifically, the key contributions are as follows:

- We identify the performance issues in existing pipeline schedules for conservative MLIPs, and we propose an instruction list to abstract the pipeline execution of conservative MLIPs.

- We propose *SymFold* to transform first-order PP schedules into a second-order instruction list, eliminating redundant recomputation and parameter replication.

- We propose *WaveK* to further reduce the pipeline bubbles with a controllable memory footprint by reordering the instruction list, and we repack micro-batches

to mitigate load imbalance.

- JanusPipe integrates with pipeline, graph, and data parallelism to enable 3D-parallel training (PP/DP/GP) of conservative MLIPs. Together, these components form a distributed training system tailored for conservative MLIP training, thereby paving the way for extending scaling laws in the MLIPs community.

## 2. Preliminaries

**MLIPs.** Modern MLIPs are typically built on graph neural networks (GNNs) composed of stacked interaction blocks, where atoms are represented as nodes and interatomic bonds as edges (Batatia et al., 2022; Deng et al., 2023; Wood et al., 2025; Fu et al., 2025). A foundational physical principle of modern MLIPs is energy conservation (Bigi et al., 2025), which requires computing forces as the negative gradient of potential energy with respect to atomic positions in the forward pass ($F = -\nabla_x E$). Although non-conservative MLIPs can directly predict forces via a readout layer (Liao et al., 2024), recent studies show that they may violate physical invariants in downstream simulations (e.g., energy/temperature drift) (Bigi et al., 2025).

**Training MLIPs with Four Phases.** Figure 1 summarizes the four-phase workflow in each training iteration: (1) **Forward Energy (FE):** We perform a forward pass to predict the total energy $E$. (2) **Forward Force (FF):** We compute atomic forces $F$ by taking the gradient of the energy $E$ w.r.t. atomic positions $x$ ($F = -\nabla_x E$), which requires the FE activations (and the corresponding computation graph) to be available. It is important to highlight that FF uses automatic differentiation to compute $\nabla_x E$ for forces, rather than back-propagating a training loss for parameter updates. To update model parameters, the iteration includes two subsequent phases that backpropagate through FF and FE. (3) **Backward Force (BF):** First, a double-backward (backward-of-backward) computation is performed with respect to the force loss $L_F$, backpropagating through the FF computation. (4) **Backward Energy (BE):** Following this, the backward computation is performed with respect to the energy loss $L_E$, backpropagating through the FE computation.

**Distributed Training Strategies.** Training large MLIP models exceeds the capacity of a single device (e.g., GPU), necessitating distributed training systems (Sriram et al., 2022; Li et al., 2025a). These include data parallelism, pipeline parallelism, and graph parallelism, which constitute orthogonal dimensions of parallelization. **Data parallelism (DP)** replicates the model and synchronizes gradients, and can be combined with state sharding (e.g., ZeRO/FSDP). Sharded parameters are materialized on demand via all-gather during computation. **Pipeline paral-**

*Table 1.* Notations.

| Symbol | Meaning |
|---|---|
| $g, x, s$ | Atomic graph, atomic positions, sequence |
| $E, F, L_E, L_F$ | Energy, force, and their respective losses |
| $h_i, \theta$ | Hidden feature of layer $i$, parameters |

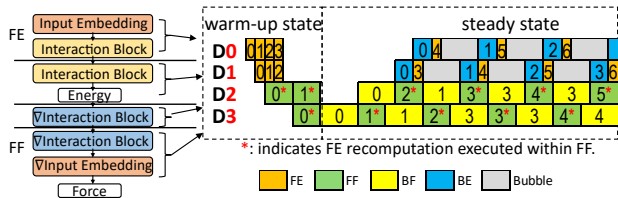

*Figure 2.* Naively applying first-order PP schedules to conservative MLIPs causes redundant FE recomputation and residual pipeline bubbles.

## 3. Observations

We present two key observations that explain why first-order PP schedules are ineffective for second-order, four-phase MLIP training.

**Observation 1: Four-phase execution causes redundant recomputation and extra memory footprint.**

FF needs to reuse the FE activations and differentiate through the FE computation graph to obtain forces. Under PP, FE and FF execute on different devices, forcing redundant recomputation to regenerate the activations and rebuild the required computation graph for FF (Figure 2). These operations lead to replicated FE parameters and activations being stored, increasing memory usage. Moreover, FE/FF share parameters, so the parameter gradients from BF and BE must be synchronized before taking an optimizer step. Redundant recomputation and parameter replication increase memory pressure and reduce end-to-end throughput.

**Observation 2: The partial order of phase execution times causes additional pipeline bubbles.**

*1) Execution time of four phases.* The execution times of the four phases on each model block obey a consistent partial order, $t_{FE} < t_{FF} < t_{BE} < t_{BF}$. Profiling results are summarized in Table 5. Specifically, FF computes forces via $F = -\nabla_x E$ and needs to compute activation gradients for backpropagation without computing parameter gradients; therefore, $t_{FF} > t_{FE}$. BE backpropagates the energy loss and updates parameters, which requires computing both parameter gradients and activation gradients, making

$t_{FF} < t_{BE}$. BF backpropagates the force loss through the force computation, and it is typically the most expensive phase because it involves double-backward.

*2) Additional bubbles.* This four-phase execution causes more bubbles in the steady state of the pipeline. As shown in Figure 2, we provide an example of training a conservative MLIP model on four devices (PP=4). D0 and D1 execute FE to produce energy in the forward pass, while D2 and D3 execute FF to compute forces. D2 and D3 execute BF, while D0 and D1 execute BE in the backward pass. In the steady state, the execution time of FE+BE on D0 and D1 differs from that of FF+BF on D2 and D3. Consequently, FE+BE on D0 and D1 cannot fully overlap FF+BF on D2 and D3, leaving an uncovered bubble of $(t_{FF}+t_{BF})-(t_{BE}+t_{FE})$ on D0 and D1 for each micro-batch. Moreover, FF recomputes FE on D2 and D3 to regenerate the required activations, which further increases pipeline bubbles in practice. Therefore, we should leverage the inherent partial order in the phase execution times to improve the overlap among the four phases.

## 4. JanusPipe

We design JanusPipe as a 3D-parallel distributed training system specifically tailored for conservative MLIPs. It represents PP execution on each device as an instruction list and incorporates two core scheduling components: SymFold and WaveK (Figures 3 and 4). SymFold transforms a first-order pipeline schedule into a second-order one by co-locating FE and FF on the same device, avoiding redundant recomputation and parameter replication. WaveK reorders the instruction lists generated by SymFold to reduce pipeline bubbles while controlling memory footprint. Additionally, JanusPipe incorporates a lightweight micro-batch repacking module, GARS (graph-aware re-scheduling), to mitigate micro-batch imbalance (details in Appendix A).

### 4.1. SymFold: Enabling Second-Order Pipeline Parallelism

We represent the four-phase MLIP training using an intermediate representation (IR) of instructions. SymFold converts

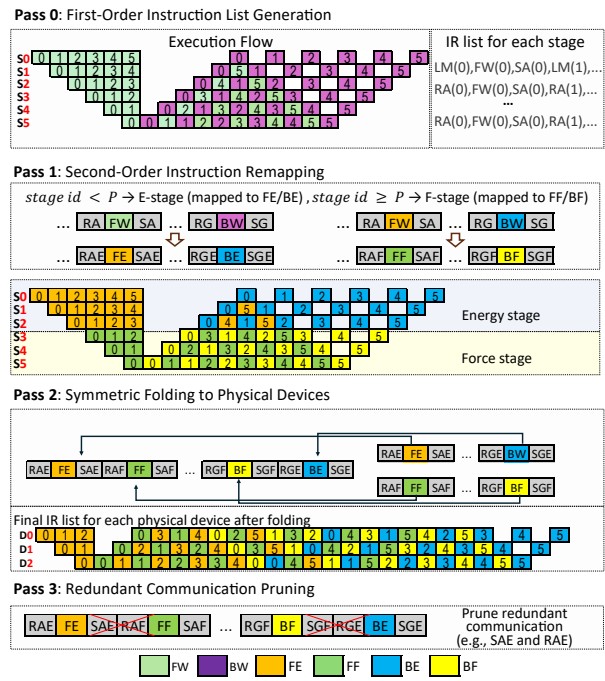

*Figure 3.* SymFold transforms a first-order PP schedule into a correct second-order schedule. For simplicity in this figure, we assume that the four phases have identical execution times.

the first-order pipeline schedule into a second-order one, ensuring training correctness through four optimization passes (i.e., passes 0–3). It places FE and FF on the same device, reusing FE's activations locally to eliminate redundancy and ensure correct gradient paths.

**Abstraction of Instruction-Level Scheduling.** We represent the pipeline schedule as an instruction list, where each device executes its own list. The instructions are shown in Table 2. The instructions should be explicitly adjusted to ensure correctness and enable efficient scheduling. We categorize instructions into three classes. (1) *Computation*: forward and backward computation, including the four MLIP phases FE/FF (forward) and BF/BE (backward, where BF corresponds to the double-backward operation). (2) *Point-to-point pipeline communication*: activation and gradient transfers between stages, including SA/RA and SG/RG, with phase-specific variants {SAE, SAF, RAE, RAF} and {SGE, SGF, RGE, RGF}. (3) *Runtime control*: data loading (LM), optimizer step (OS), and data-parallel synchronization (AR). With this device-independent abstraction, we can model MLIP pipeline execution without relying on physical device mappings.

**Pass 0: First-Order Instruction List Generation.** As shown in Figure 3, given the pipeline degree $P$ and the number of micro-batches $N_{\text{mb}}$, we generate a first-order pipeline schedule as a uniform instruction list with $2P$ virtual stages.

*Table 2.* Abstracted instructions.

| Cat. | Instr. | Definition |
| --- | --- | --- |
| FW | FE | Forward Energy Pass |
| | FF | Forward Force Pass |
| BW | BE | Backward Energy Pass |
| | BF | Backward Force Pass |
| SA | SAE | Send Activation for Energy |
| | SAF | Send Activation for Force |
| RA | RAE | Receive Activation for Energy |
| | RAF | Receive Activation for Force |
| SG | SGE | Send Gradient for Energy |
| | SGF | Send Gradient for Force |
| RG | RGE | Receive Gradient for Energy |
| | RGF | Receive Gradient for Force |
| OS | OS | Optimizer Step |
| LM | LM | Load Micro-batch |
| AR | AR | All-Reduce for DP |

It provides finer-grained partitioning, allowing the model to be divided into $2P$ stages instead of $P$. By doubling the stages, we simplify MLIP-specific optimizations such as labeling FW and BW steps as energy or force and co-locating them on the same device.

**Pass 1: Second-Order Instruction Remapping.** Pass 1 transforms first-order instructions into second-order ones. The remapping follows two rules: **Rule 1.** *Second-order compatibility*: The original first-order instructions are transformed into the instructions listed in Table 2, so as to be compatible with second-order MLIP training. For each micro-batch, we assign the first half of forward instructions (FW, SA, RA) to *forward energy* (FE, SAE, RAE), and the second half to *forward force* (FF, SAF, RAF). Similarly, backward instructions (BW, SG, RG) are remapped to *backward energy* (BE, SGE, RGE) and *backward force* (BF, SGF, RGF). **Rule 2.** *Gradient-path correctness*: The remapping must ensure that the final gradient update remains mathematically equivalent to the original computation, guaranteeing training correctness. The total loss is $L_{\text{total}} = L_E + L_F$ (symbols are defined in Table 1), and the full gradient is:

$$\frac{\partial L_{\text{total}}}{\partial \theta} = \underbrace{\frac{\partial L_E}{\partial h_i} \cdot \frac{\partial h_i}{\partial \theta}}_{\text{BE: first-order term}} + \underbrace{\frac{\partial L_F}{\partial h_i} \cdot \frac{\partial h_i}{\partial \theta}}_{\text{BF: first-order term}} + \underbrace{\frac{\partial L_F}{\partial \left(\frac{\partial E}{\partial h_i}\right)} \cdot \frac{\partial^2 E}{\partial h_i \partial \theta}}_{\text{BF: second-order term}}$$

(1)

Specifically, we move the first-order gradient term induced by $L_F$ into BE. The remaining second-order term is then assigned exclusively to BF:

$$\frac{\partial L_{\text{total}}}{\partial \theta} = \underbrace{\left(\frac{\partial L_E}{\partial h_i} + \frac{\partial L_F}{\partial h_i}\right) \cdot \frac{\partial h_i}{\partial \theta}}_{\text{BE: merged first-order term}} + \underbrace{\frac{\partial L_F}{\partial \left(\frac{\partial E}{\partial h_i}\right)} \cdot \frac{\partial^2 E}{\partial h_i \partial \theta}}_{\text{BF: second-order term}}$$

(2)

BE accumulates all first-order contributions, while BF processes the second-order term.

**Pass 2: Symmetric Folding to Physical Devices.** Pass 2 symmetrically folds energy and force instructions onto the same device. Specifically, we fold the $2P$ virtual stages onto $P$ physical devices with symmetric alignment along the time axis. To preserve execution order, in the forward flow, energy instructions (FE, SAE, RAE) precede force instructions (FF, SAF, RAF) on each device. In the backward flow, backward force instructions (BF, SGF, RGF) precede backward energy instructions (BE, SGE, RGE). We then symmetrically fold them so that virtual stage $S_i$ and its paired virtual stage $S_{2P-1-i}$ are co-located on the same device. This symmetric folding mapping helps FF reuse the activations generated by FE.

$$M(s_v) = \begin{cases} s_v & \text{if } 0 \leq s_v < P \\ 2P - 1 - s_v & \text{if } P \leq s_v < 2P \end{cases} \quad (3)$$

**Pass 3: Redundant Communication Pruning.** Following Pass 2, when the source and destination stages of communication instructions are folded onto the same device, redundant communication instructions can be removed. Specifically, this placement results in adjacent virtual stages being co-located, so their send/receive instructions are reduced to intra-device data transfers. For example, in Figure 3, Stage 2 (S2) originally needs to send activations to Stage 3 (S3), and S3 in turn sends gradients back to S2, but once S2 and S3 are co-located on the same device (D2), these instructions are no longer needed. Pass 3 thus prunes these redundant send/receive instructions.

### 4.2. WaveK: Adaptive Pipeline Scheduling for Imbalanced Four-Phase Workloads

WaveK takes the SymFold schedule as input and improves phase overlap to reduce the extra pipeline bubbles described in Section 3. WaveK leverages the consistent partial order among phases, which is observed across conservative MLIP models as $t_{\text{FE}} < t_{\text{FF}} < t_{\text{BE}} < t_{\text{BF}}$ (see Appendix B.2). As shown in Figure 4, we visualize the instruction lists after pass 3 as a timeline, and observe that energy stages (FE/BE) overlap with force stages (FF/BF), though additional bubbles remain in the steady state. WaveK addresses this by reorganizing the SymFold instruction list into WaveK units, each of which groups $k$ micro-batches as a scheduling block. Each unit contains a forward wave (WaveK-F) that executes FE and FF for $k$ micro-batches, and a backward wave (WaveK-B) that executes BF and BE for the same $k$ micro-batches. As a result, WaveK overlaps adjacent unit boundaries, eliminating the residual bubbles of the previous unit and the initial bubbles of the next unit. The unit size $k$ controls the tradeoff between throughput and memory and is selected through an offline search with a controllable memory footprint.

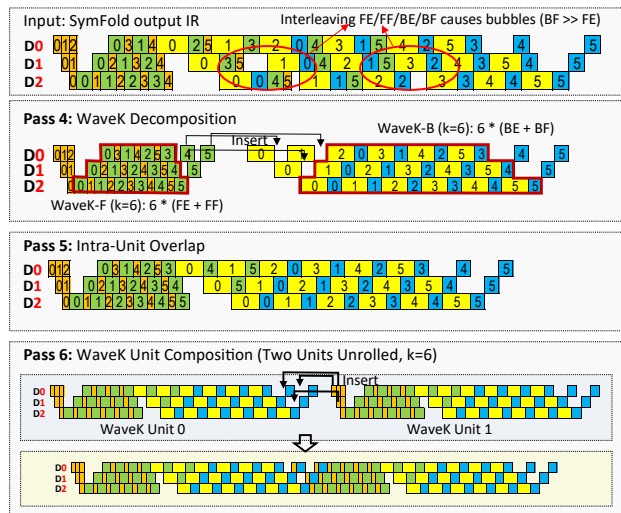

*Figure 4.* WaveK organizes the instructions into WaveK units and overlaps unit boundaries to reduce pipeline bubbles under the four-phase partial order.

**Pass 4: WaveK Decomposition.** Pass 4 takes the Sym-Fold schedule as input and decomposes the four-phase execution into two parts: WaveK-F and WaveK-B. As shown in Figure 4, WaveK-F contains forward (FE, FF) phases for $k$ micro-batches and achieves a steady state without pipeline bubbles. WaveK-B contains the backward (BF, BE) phases and their corresponding instructions, and it similarly achieves a steady state without bubbles. As a result, WaveK-F and WaveK-B are bubble-free in steady state, while the remaining bubbles are confined to the WaveK-F/WaveK-B boundary and unit boundaries, allowing overlap.

**Pass 5: Intra-Unit Overlap.** At the boundaries of WaveK-F/WaveK-B, the pipeline is not yet fully overlapped. At the end of WaveK-F, the pipeline has residual bubbles dominated by FF. At the beginning of WaveK-B, the pipeline has not yet reached steady state, creating initial bubbles dominated by BE. To mitigate these boundary bubbles, WaveK overlaps the two waves within a WaveK unit by inserting FF instructions from the WaveK-F residual into the initial bubbles of WaveK-B.

**Pass 6: WaveK Unit Composition.** After intra-unit boundary overlap, residual bubbles may still remain at the boundaries between consecutive WaveK units, requiring further mitigation. WaveK combines adjacent units by overlapping their execution across unit boundaries. As shown in Figure 4 (Pass 6), WaveK fills the residual bubbles at the end of unit 0 and the initial bubbles at the beginning of unit 1 with useful execution from the neighboring unit. Across units, the remaining residual bubble of size $t_{\text{BE}}$ can host FE operations from the next unit. Figure 5 illustrates two different values of $k$ ($k = 4$ and $k = 6$) with $N_{\text{mb}} = 12$

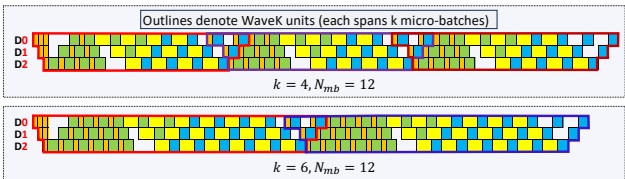

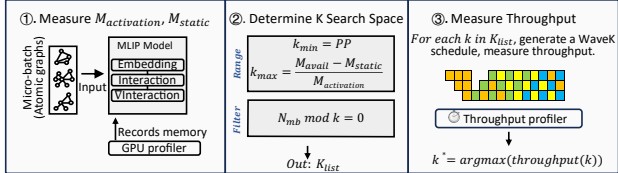

*Figure 5.* WaveK schedules with different $k$ (fixed $N_{\text{mb}}$=12). Top: $k$=4. Bottom: $k$=6.

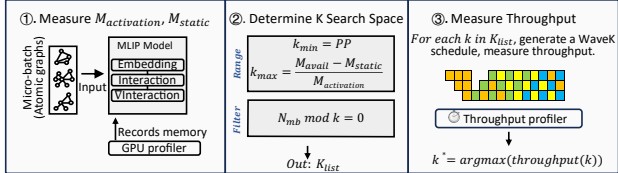

*Figure 6.* Offline tuning selects the WaveK unit size $k$ under a memory constraint.

micro-batches. The top shows the case of $k = 4$ with three WaveK units, and the bottom shows the case of $k = 6$ with two WaveK units, resulting in fewer pipeline bubbles and achieving higher throughput. Increasing $k$ reduces the number of unit boundaries, thereby confining bubbles to unit boundaries and improving steady-state overlap.

**Bubble Analysis.** We analyze pipeline bubbles at three levels. **Intra-unit bubbles.** Within a WaveK unit, the boundary between WaveK-F and WaveK-B creates a residual bubble of size $(t_{\text{BE}} - t_{\text{FF}})$. With pipeline degree $P$, micro-batch number $N_{\text{mb}}$, and $\frac{N_{\text{mb}}}{k}$ WaveK units, the total intra-unit bubble size is $(t_{\text{BE}} - t_{\text{FF}}) \cdot P \cdot \frac{N_{\text{mb}}}{k}$. **Inter-unit bubbles.** Across adjacent WaveK units, the remaining boundary creates a residual bubble of size $(t_{\text{BF}} - t_{\text{FE}})$. Therefore, the total inter-unit bubble size is $(t_{\text{BF}} - t_{\text{FE}}) \cdot P \cdot \left(\frac{N_{\text{mb}}}{k} - 1\right)$. **Effect of stage doubling.** By doubling the pipeline stages (from $P$ to $2P$), SymFold facilitates finer-grained partitioning. Under approximately uniform partitioning, the per-stage phase times scale down as $t_\phi^{(2P)} \approx t_\phi^{(P)}/2$ for $\phi \in \{\text{FE}, \text{FF}, \text{BF}, \text{BE}\}$. Hence the steady-state bubble term is reduced by $\approx 2\times$: $B \approx (t_{\text{BF}} + t_{\text{BE}} - t_{\text{FF}} - t_{\text{FE}}) \cdot \frac{P \cdot N_{\text{mb}}}{2k} - \frac{P}{2}(t_{\text{BF}} - t_{\text{FE}})$, typically $k \geq P$.

WaveK exposes a trade-off between throughput and memory controlled by the unit size $k$. More specifically, a larger $k$ enables deeper overlap and reduces bubbles, but it also increases the number of in-flight micro-batches whose forward activations must be retained until BE completes.

**Offline Tuning under Memory Constraints.** To determine the optimal $k$, we employ an offline tuning approach (Figure 6) that explores feasible $k$ values under memory constraints and selects the **value of** $k$ that maximizes throughput.

*Step 1. Measure Memory Footprint.* Given a specific model configuration and micro-batch size, we measure the following under the target parallel configuration: (1) Static memory $M_{\text{static}}$ (parameters, gradients, and optimizer states). (2) Activation memory $M_{\text{activation}}$. We set an effective memory budget $M_{\text{mem}} = M_{\text{GPU}} - M_{\text{reserve}}$. We profile a worst-case micro-batch $MB_{\text{max}}$ by packing the largest graphs (high atoms/edges) up to the atom budget $C_{\text{max}}$ (e.g., 400 atoms), and measure $M_{\text{activation}}(MB_{\text{max}}, k)$ as the peak activation memory in one four-phase iteration. We choose the largest $k$ such that $M_{\text{static}} + M_{\text{activation}}(MB_{\text{max}}, k) \leq M_{\text{mem}}$. We reserve a small safety buffer $M_{\text{reserve}}$ to avoid OOM due to allocator fragmentation, and discard candidate values of $k$ that exceed the per-rank memory budget during offline tuning.

*Step 2. Determine Search Space.* Feasible $k$ values must lie within the range defined by $k_{\text{max}} = \left\lfloor \frac{M_{\text{mem}} - M_{\text{static}}}{M_{\text{activation}}} \right\rfloor$, $k_{\text{min}} = P$. We prefer $k$ values that divide $N_{\text{mb}}$ to avoid leftover micro-batches that cannot fully fill the steady state. We record these values in $K_{\text{list}}$.

*Step 3. Profile Throughput and Select.* For each $k$ in $K_{\text{list}}$, we generate the WaveK PP schedule and measure the average throughput over five training steps to obtain $k^*$: $k^* = \arg\max_{k \in K_{\text{list}}} \text{Throughput}(k)$.

This offline search is performed once before training (a reasonable default is $k$=$P$ when tuning is skipped) and produces a throughput-optimized schedule with controllable memory footprint.

## 5. Evaluation

### 5.1. Experimental Setup

**Datasets.** We evaluate on a mixed dataset of ODAC23 (Sriram et al., 2023), OMat24 (Barroso-Luque et al., 2024), and OMol25 (Levine et al., 2025), sampling each dataset with equal probability. Each training iteration processes a global batch of 12,800 atoms, split into micro-batches with a fixed size of 400 atoms. **Models.** We evaluate two representative MLIP families with distinct computation and memory behavior. UMA (Wood et al., 2025) adopts a sparse Mixture-of-Linear-Experts (MoLE) design, where expert weight matrices are combined via weighted averaging into a temporary linear transform. This design yields parameter sparsity, retains dense computation, and reduces activation memory. eSEN (Fu et al., 2025) is fully dense. We evaluate four models (UMA-1.2B/2.3B and eSEN-100M/220M). Specifically, their hyperparameters are provided in Appendix B (Table 4). **Hardware and software.** All experiments are conducted on a cluster with ARMv8 CPUs and NVIDIA A100-40GB GPUs, using CUDA 12.4 and PyTorch 2.6. **Parallel configuration.** We vary the pipeline, graph, and data parallelism dimensions, denoted as $P$, $G$, and $D$. **Baselines.** Since no existing pipeline schedules support second-order

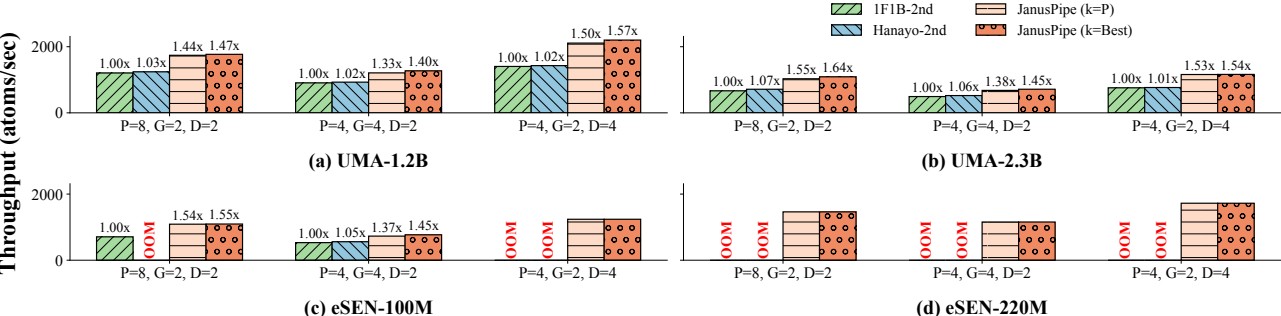

*Figure 7.* End-to-end throughput of 1F1B-2nd, Hanayo-2nd, and JanusPipe across MLIP models and PP/GP/DP settings.

training, we adopt two widely used first-order baselines and adapt them accordingly for second-order training. *1F1B-2nd* is based on Megatron-LM's 1F1B pipeline schedule, extended to support second-order training (Narayanan et al., 2021). *Hanayo-2nd* adopts the wave-style schedule from Hanayo (Liu et al., 2023) (rooted in Chimera (Li & Hoefler, 2021) and used in DeepSeek DualPipeV (Guo et al., 2025)). In these baselines, FE is recomputed locally during FF for correct backpropagation. Although Hanayo supports multiple waves ($W>1$), MLIPs are shallow (10–20 layers due to over-smoothing (Zhao & Akoglu, 2020)), making $S=2WP$ infeasible; thus we use a single wave ($W=1$, $S=2P$) for fairness. Both baselines form micro-batches via greedy packing. Additionally, JanusPipe has two configurations of $k$, the number of micro-batches per WaveK unit: $k=P$ (minimal-wave configuration, setting $k$ to the pipeline degree $P$) and $k=$Best (selecting the optimal $k$ under device memory constraints through offline tuning). Unless specified, all JanusPipe results use the default configuration with GARS enabled; baselines exclude SymFold/WaveK/GARS and keep their original scheduling logic unchanged. **Metrics.** We report training throughput as *atoms processed per second (atoms/sec)*. All evaluation runs use a 10-iteration warm-up, followed by 100 consecutive training iterations.

## 5.2. Overall Performance

**End-to-End Performance** As shown in Figure 7, Janus-Pipe achieves superior end-to-end throughput across all models and parallel settings. On average, JanusPipe ($k=$Best) delivers $1.51\times$ and $1.45\times$ higher throughput than 1F1B-2nd and Hanayo-2nd, respectively. These gains stem from better utilization of the four-phase (double-backward) execution by reducing redundant recomputation and improving phase overlap. SymFold co-locates FE and FF to reuse FE-generated activations and avoid recomputing FE across stages, while WaveK organizes execution into WaveK units to improve overlap among FE/FF/BF/BE. Compared with $k=P$, $k=$Best uses the available memory budget to form larger WaveK units with fewer unit boundaries, which typically increases overlap and improves throughput. All meth-

ods run successfully on UMA-1.2B/2.3B, and JanusPipe consistently outperforms the baselines. For eSEN, Janus-Pipe avoids the OOM failures of the baselines on eSEN-220M via a finer-grained pipeline partition (SymFold), and $G=4$ further prevents OOM on eSEN-100M by reducing per-rank activation memory through graph splitting.

**Memory Efficiency** As shown in Figure 8, JanusPipe ($k=$Best) achieves higher throughput by utilizing memory close to the device limit, while JanusPipe ($k=P$) minimizes memory footprint yet still outperforms 1F1B-2nd. Janus-Pipe ($k=P$) reduces peak GPU memory by up to 20.56% and 42.70% compared to 1F1B-2nd and Hanayo-2nd, respectively. We also observe structural memory asymmetry (Stage 0, which embeds rich atom features, consistently shows higher peak memory) and per-micro-batch memory-usage fluctuations from heterogeneous graphs. These fluctuations arise because graphs vary in atom counts; even with a fixed atom budget per micro-batch, the atom-count distribution across micro-batches remains imbalanced. Hanayo partitions the model into finer-grained stages and adopts a wave-shaped pipeline schedule to reduce bubbles. However, this schedule prolongs the lifetime of per-micro-batch activations in the pipeline, especially for four-phase MLIP training. As a result, more activations remain live simultaneously, increasing peak memory compared to 1F1B-2nd and JanusPipe.

## 5.3. Ablation Study

We start from 1F1B-2nd and progressively enable SymFold and WaveK. SymFold improves throughput by up to 23% by enabling a finer-grained model partition (doubling the number of stages) and co-locating FE and FF, reusing activations and reducing cross-device synchronization. WaveK adds a further 18% speedup by improving overlap among the four phases. In total, enabling all components yields the highest throughput, consistent with the end-to-end gains. We report more details of JanusPipe and ablations for all components (GARS adds ∼11% on average) in Appendix B.3.

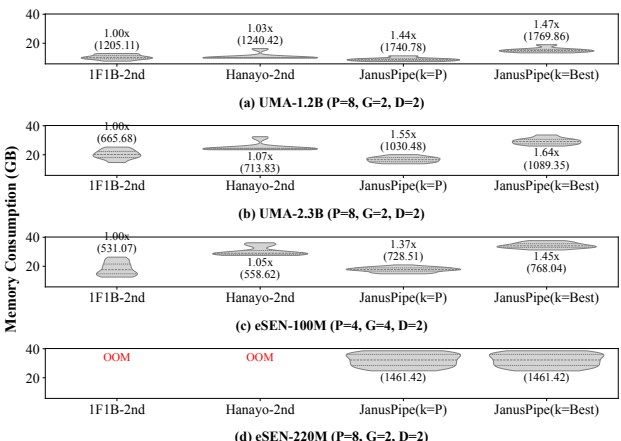

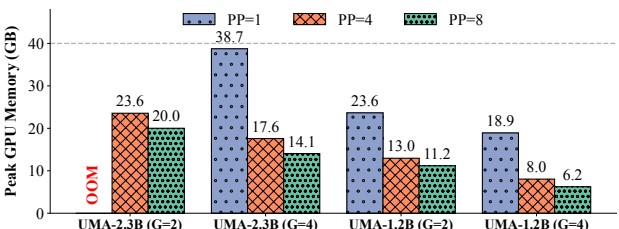

*Figure 8.* Peak device memory across 32 GPUs (violin plots), with absolute throughput (atoms/sec) and relative speedup annotated above each violin.

![Figure 9 bar chart]

*Figure 9.* Peak GPU memory versus pipeline degree $P$ on UMA-1.2B and UMA-2.3B under $G \in \{2, 4\}$.

### 5.4. WaveK Sensitivity

The WaveK unit size $k$ reveals a clear throughput–memory trade-off. Initially, throughput increases with larger $k$ because pipeline bubbles decrease, but larger $k$ is constrained by device memory. This motivates selecting $k$ under a memory budget. On UMA-1.2B ($P=4, G=D=1$), $k=8$ achieves the largest speedup of $1.05\times$ over the $k=4$ baseline.

### 5.5. Scalability Analysis

We evaluate scalability by scaling the PP dimension from $P=4$ to $P=16$ (from 8 to 32 GPUs), while fixing GP and DP to $G=2, D=1$. JanusPipe achieves 70% strong scaling efficiency and 94% weak scaling efficiency on UMA-2.3B. These results indicate that JanusPipe's schedule design remains effective as we increase pipeline depth.

### 5.6. Peak GPU Memory under Pipeline Parallelism

We characterize peak GPU memory under PP. Using UMA-1.2B and UMA-2.3B, we sweep the pipeline degree $P \in \{1, 4, 8\}$ under two GP configurations ($G \in \{2, 4\}$) and report the peak GPU memory (maximum across devices); OOM denotes exceeding the 40 GB upper limit. Figure 9 shows that increasing $P$ consistently reduces peak memory

and can convert OOM cases into runnable ones. For UMA-2.3B with $G=4$, the peak memory decreases from 38.74 GB at $P=1$ to 12.84 GB at $P=4$ and 9.24 GB at $P=8$. For UMA-2.3B with $G=2$, $P=1$ is OOM, whereas both $P=4$ and $P=8$ complete.

### 5.7. Additional Results

Additional results (ablation, scaling, bubble breakdown, correctness, and micro-benchmarks) are provided in Appendix B.

## 6. Related Works

**Pipeline parallelism for first-order training.** Pipeline parallelism (PP) has been widely studied for first-order training workloads with a two-phase dependency (forward then backward), exemplified by Megatron-LM's 1F1B schedule (Narayanan et al., 2021) and subsequent schedule variants such as Hanayo and DualPipe (Liu et al., 2023; Guo et al., 2025). Recent systems further automate pipeline scheduling via search or optimization (Zheng et al., 2022; Li et al., 2025b) and improve memory control for LLM training (Wan et al., 2025). However, their scheduling IR usually assumes a two-phase forward/backward dependency but does not model the four-phase execution of MLIPs. **Training large-scale MLIPs.** Large-scale MLIPs increasingly use data and graph parallelism for training (Sriram et al., 2022; Wood et al., 2025). Complementary to training, DistMLIP applies graph parallelism to multi-GPU MLIP inference (Han et al., 2025). Prior work has scaled *non-conservative* MLIPs using standard first-order distributed training (Li et al., 2025a); however, these approaches do not apply to *conservative* (double-backward) MLIPs. **Complementary techniques.** JanusPipe targets pipeline parallelism scheduling for conservative MLIPs and is orthogonal to several existing optimizations. In particular, it can be combined with kernel-level accelerations (e.g., FlashTP) (Lee et al., 2025), compiler/kernel-based MLIP acceleration stacks (e.g., NequIP/Allegro) (Tan et al., 2025), and data-parallel memory optimizations based on state sharding (e.g., ZeRO and FSDP) (Rajbhandari et al., 2020; Zhao et al., 2023). We do not include these techniques in this paper, but they are compatible with JanusPipe.

## 7. Conclusion

In this paper, we presented JanusPipe, which addresses two key inefficiencies in distributed training of conservative MLIPs that require double-backward execution. It introduces an instruction-level schedule abstraction for four-phase execution. SymFold enforces FE–FF co-location to eliminate redundant recomputation, while WaveK reorganizes execution under memory constraints to reduce pipeline

bubbles; we further apply micro-batch repacking to mitigate load imbalance. On UMA and eSEN with 32 GPUs, JanusPipe improves throughput by $1.51\times$ and $1.45\times$ on average over baselines, and enables previously OOM configurations under the same device memory budget. JanusPipe integrates pipeline, data, and graph parallelism to support 3D-parallel distributed training of MLIPs. In the future, we hope JanusPipe will lower the barrier to distributed training of large-scale MLIPs and inspire further research and engineering on MLIP training infrastructure, helping pave the way for future scaling-law studies in the MLIP community.

## Acknowledgments

The authors are grateful to the Area Chairs and the anonymous reviewers for their constructive comments. This work is supported by the following funding: National Science Foundation of China (T2125013, 92270206, 62372435, 62502501), Beijing Natural Science Foundation (4254087), and the Innovation Funding of ICT, CAS. Part of the training is performed on the robotic AI-Scientist platform of Chinese Academy of Sciences.

## Impact Statement

This paper presents work whose goal is to advance the field of Machine Learning. There are many potential societal consequences of our work, none which we feel must be specifically highlighted here.

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

# A. Graph-Aware Re-Scheduling (GARS)

GARS is an important module that repacks atomic graphs into micro-batches to mitigate load imbalance induced by a long-tailed graph size distribution. Real-world molecular and materials datasets exhibit highly skewed graph size distributions (Table 3), which directly translate to computational imbalance across all parallelism dimensions (PP/DP/GP). Without proper handling, this imbalance creates performance bottlenecks. While our main design targets distributed training for MLIPs, heterogeneous per-graph costs under multi-dimensional parallelism (PP/GP/DP) can create persistent stragglers, amplifying pipeline bubbles and communication stalls.

## A.1. Motivation

**Long-tailed graph sizes.** Atomic graphs in MLIP datasets exhibit substantial size skew, ranging from a few atoms to hundreds (or more) atoms per graph (Table 3). Even when we enforce a fixed atom budget per micro-batch, greedy sequential packing can still yield high runtime variance across micro-batches. Consequently, under multi-dimensional distributed parallelism, per-rank runtime can still vary because graph sizes (atoms/edges) of micro-batch are unevenly mixed.

*Table 3.* Statistics of atomic graph sizes in representative datasets.

| Dataset | Count | Mean | P50 | P90 | P99 | Max |
|---|---|---|---|---|---|---|
| OMat24 (Barroso-Luque et al., 2024) [*] | 11,388,510 | 14 | 14 | 18 | 30 | 184 |
| OMol25 (Levine et al., 2025) | 101,666,280 | 52 | 38 | 114 | 202 | 350 |
| ODAC23 (Sriram et al., 2023) | 35,871,295 | 202 | 175 | 355 | 537 | 905 |
| Mixed | 148,926,085 | 85 | 53 | 213 | 427 | 905 |

[*] This work uses the `rattled-1000` subset of OMat24.

**Impact under PP/GP/DP.** This heterogeneity affects distributed execution in three ways. Figure 10 illustrates how micro-batch heterogeneity creates PP bubbles and DP synchronization stalls. (i) **DP imbalance:** DP ranks assigned micro-batches containing larger graphs run longer, causing faster ranks to wait at parameter gradient synchronization before each optimizer step. (ii) **PP bubbles:** uneven micro-batch compute times create stage-level imbalance. A heavier micro-batch delays its stage and propagates stalls to downstream stages, leaving some GPUs idle (pipeline bubbles). For example, if $MB_1$ is heavier than $MB_0$, stage 0 may finish $MB_0$ early but take longer on $MB_1$; stage 1 can process $MB_0$ but must then wait for activations from stage 0 for $MB_1$, creating an idle interval on stage 1. Figure 10 visualizes the resulting pipeline bubbles under PP and the straggler-induced waiting under DP. (iii) **GP inefficiency:** GP may split or balance graphs based on node count (atom count), so each GP rank receives a similar number of nodes. However, most MLIP operations are edge-centric, and the runtime is largely determined by the number of edges, leading to suboptimal balance and unnecessary communication. In MLIPs, the execution time is closely related to the number of edges because interaction modeling and feature passing are performed over edges. As a result, even small graphs can still trigger halo All-Gather at each interaction layer, increasing communication overhead.

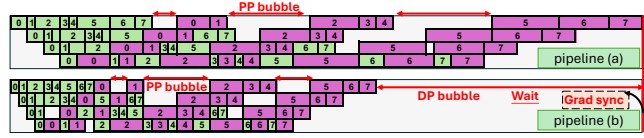

*Figure 10.* Impact of micro-batch heterogeneity under PP and DP: bubbles and synchronization stalls.

## A.2. Lightweight Solver: Heuristic Algorithm

GARS reduces step-time variance by repacking graphs into better-balanced micro-batches and tagging each micro-batch to select an efficient GP execution mode: *comm-free* local execution for small-graph micro-batches, and *dist* execution that splits oversized graphs across GP ranks when necessary. Therefore, a micro-batch that contains only small graphs can be executed locally without splitting across GP ranks, avoiding halo communication. Since parameter updates are applied at the global-batch granularity, reordering graphs within a global batch does not change the training semantics, but can produce more balanced micro-batches. We present a lightweight solver for graph-aware rescheduling: a heuristic that reorders graphs within each global batch to reduce imbalance across micro-batches.

Given a global batch $B = \{g_1, \ldots, g_M\}$, GARS partitions it into $N_{\mathrm{mb}}$ micro-batches. It uses atom count $\mathrm{size}(g)$ as a lightweight proxy for graph cost and applies a *pack-and-shuffle* heuristic (Algorithm 1) with three steps.

**(1) Inter-micro-batch packing for load balance.** GARS first sorts graphs by atom count (line 3) and assigns each graph to the micro-batch with the current minimum total size (line 6). This reduces micro-batch imbalance caused by long-tailed graph sizes, which directly lowers step-time variance under PP/DP synchronization.

**(2) Intra-micro-batch shuffling for GP balance.** After step 1, graphs within each micro-batch tend to follow a size order (large→small). Under GP, this ordering can skew per-rank workloads by placing multiple large graphs on the same GP ranks. Since MLIP interaction blocks are dominated by edge-wise operations, and edge count grows superlinearly with atom count, such skew can create GP stragglers. To mitigate this, GARS shuffles the graph order within each micro-batch (line 10) before GP bin assignment, promoting a heterogeneous mix of graph sizes per rank.

**(3) Type tagging: comm-free vs. dist.** GARS then determines the execution mode of each micro-batch (lines 13–15). If the largest graph fits within the per-rank atom budget, i.e., $\max_{g \in MB_j} \mathrm{size}(g) \le C_{\mathrm{rank}}(MB_j)$, we tag $MB_j$ as *comm-free* (line 13), meaning graphs can be kept local and halo All-Gather is avoided. Otherwise, we tag it as *dist* (line 15), where oversized graphs are handled with distributed execution and halo All-Gather. This rule captures the trade-off between avoiding redundant halo communication for small graphs and enabling distributed execution for large graphs. For *comm-free* micro-batches, GARS optionally applies a second-level min-load assignment to map graphs to $d_{\mathrm{gp}}$ local bins, further balancing per-rank compute.

---

**Algorithm 1** GARS: Lightweight Pack-and-Shuffle Algorithm.

---

1: **Input:** Global batch $B = \{g_1, \ldots, g_M\}$; number of micro-batches $N_{\mathrm{mb}}$; GP degree $d_{\mathrm{gp}}$
2: **Output:** $\{MB_1, \ldots, MB_{N_{\mathrm{mb}}}\}$ with type tags $\{\mathsf{comm\_free}, \mathsf{dist}\}$
3: Sort $B$ in descending order by $\mathrm{size}(g)$
4: Initialize $MB\_list \leftarrow \{\emptyset, \ldots, \emptyset\}$ of length $N_{\mathrm{mb}}$
5: **for** $i = 1, \ldots, M$ **do**
6:     $j^\star \leftarrow \arg\min_j \sum_{g \in MB_j} \mathrm{size}(g)$
7:     $MB_{j^\star} \leftarrow MB_{j^\star} \cup \{g_i\}$
8: **end for**
9: **for** $j = 1, \ldots, N_{\mathrm{mb}}$ **do**
10:     $\mathrm{SHUFFLE}(MB_j)$
11:     $C_{\mathrm{rank}}(MB_j) \leftarrow \dfrac{\sum_{g \in MB_j} \mathrm{size}(g)}{d_{\mathrm{gp}}}$
12:     **if** $\max_{g \in MB_j} \mathrm{size}(g) \le C_{\mathrm{rank}}(MB_j)$ **then**
13:         $\mathrm{type}(MB_j) \leftarrow \mathsf{comm\_free}$
14:     **else**
15:         $\mathrm{type}(MB_j) \leftarrow \mathsf{dist}$
16:     **end if**
17: **end for**

---

### A.3. Complexity and Correctness

**Complexity.** The overall complexity is $O(M \log M + M \log N_{\mathrm{mb}})$ (sorting and repeated min-load placement), which is negligible compared to GPU training. **Correctness.** GARS only changes the ordering and grouping of graphs within a global batch; it does not modify per-graph computations. The training objective is a sum over per-graph losses, so the step gradient depends on the set of graphs rather than their partition into micro-batches. Therefore, repacking preserves the mathematical gradient.

## B. Experimental Details

### B.1. Experimental Setup

**Second-order adaptation.** For each micro-batch, we expand a first-order PP schedule into a four-phase instruction sequence (FE/FF/BF/BE) by mapping each forward block to {FE,FF} and each backward block to {BF,BE}, while enforcing FE→FF→BF→BE dependencies. The baseline PP schedule ignores double-backward four-phase dependencies, so FF cannot directly reuse FE activations. PyTorch does not provide a practical mechanism to serialize and transfer autograd graph state (e.g., grad_fn and saved tensors) across processes. Therefore, we follow a first-order PP adaptation: we

recompute FE on the FF side to recreate the required computation graph and activations when needed. Moreover, FF-side stages keep replicated parameters and synchronize parameter gradients before the optimizer step. Hanayo and related PP schedulers are designed for two-phase (forward/backward) first-order training and do not directly model the four-phase double-backward dependencies. In addition, conservative MLIPs are typically shallow (10–20 layers), as deeper GNNs suffer from over-smoothing that degrades prediction accuracy. Consequently, Hanayo configurations with $W > 1$ are difficult to realize in practice, because they require a large number of pipeline stages (e.g., $S = 2WP$) to be effective. To ensure correctness, we **re-implement** and extend the computation/communication instruction set to support double-backward execution, instead of reusing the original two-phase runtime. For example, BF involves higher-order gradient propagation through message-passing interaction intermediates, and thus we must carefully define which higher-order gradients are passed across stages and specify dedicated computation instructions (FF/BF) to ensure correctness.

**Model Configurations** Table 4 summarizes the evaluated MLIP model configurations. Here $L_{\max}$ and $M_{\max}$ denote the maximum spherical-harmonic degree and order used in the equivariant representation, following the UMA/eSEN default settings. MoLE Experts indicates whether the MoE layer is enabled; "Dense" means no experts (i.e., a standard dense MLP), while a number denotes the expert count. We scale eSEN by adjusting width and depth; eSEN-220M is wider but shallower than eSEN-100M under our configuration.

*Table 4.* Model configurations for evaluation.

| Model | # Radial basis | $N_{\text{channel}}$ | # Layers | $L_{\max}$ | $M_{\max}$ | MoLE experts |
|---|---|---|---|---|---|---|
| UMA-1.2B | 64 | 128 | 8 | 2 | 2 | 128 |
| UMA-2.3B | 64 | 128 | 16 | 2 | 2 | 128 |
| eSEN-100M | 64 | 256 | 16 | 2 | 2 | Dense |
| eSEN-220M | 64 | 512 | 8 | 2 | 2 | Dense |

**Hardware and Software** All experiments are conducted on a cluster with 8 nodes, each equipped with two 64-core ARMv8 CPUs (Kunpeng 920) and 4 NVIDIA A100-40G-PCIe GPUs, running Driver 535.104.12, CUDA 12.4, and PyTorch 2.6.

### B.2. Four Phase Profiler

We observe that the absolute and relative execution times of the four phases vary across models, but the timing relationships follow a consistent partial order. As shown in Table 5, the measured phase times consistently satisfy $t_{\text{FE}} < t_{\text{FF}} < t_{\text{BE}} < t_{\text{BF}}$. While the exact ratios are model-dependent, this partial order holds across all evaluated models and underpins the design of WaveK.

**Theoretical Foundation of Partial Order.** The observed partial order is not merely empirical but grounded in the computational characteristics of each phase:

- $t_{\text{FE}} < t_{\text{FF}}$: FF computes the gradient of energy with respect to atomic positions ($F = -\nabla_x E$), which requires backward propagation through the FE computation graph to obtain activation gradients. While FF does not compute parameter gradients, it must compute activation gradients, making it computationally more expensive than FE's forward pass.

- $t_{\text{FF}} < t_{\text{BE}}$: BE backpropagates the energy loss and computes both activation gradients and parameter gradients. In contrast, FF computes only activation gradients (no parameter gradients), so we have $t_{\text{FF}} < t_{\text{BE}}$.

- $t_{\text{BE}} < t_{\text{BF}}$: BF performs double-backward computation by backpropagating through the FF phase, whereas BE backpropagates through the FE phase. Since FF is typically more expensive than FE ($t_{\text{FF}} > t_{\text{FE}}$), the backward pass through FF (BF) tends to be costlier than the backward pass through FE (BE). Therefore, we have $t_{\text{BE}} < t_{\text{BF}}$.

This computational analysis shows that the partial order $t_{\text{FE}} < t_{\text{FF}} < t_{\text{BE}} < t_{\text{BF}}$ is inherent to conservative MLIPs and is expected to hold across different model architectures.

*Table 5.* Per-micro-batch compute time of FE/FF/BE/BF under $P=8$, $G=2$, $D=2$ (each row normalized by its FE time; FE=1×).

| Model | FE (ms) | FF (ms) | BE (ms) | BF (ms) |
|---|---|---|---|---|
| eSEN-220M | 24.96 (1.00×) | 57.67 (2.31×) | 64.30 (2.58×) | 118.92 (4.76×) |
| eSEN-100M | 52.98 (1.00×) | 84.88 (1.60×) | 85.29 (1.61×) | 175.67 (3.32×) |
| UMA-2.3B | 58.41 (1.00×) | 87.22 (1.49×) | 98.15 (1.68×) | 190.73 (3.27×) |
| UMA-1.2B | 26.25 (1.00×) | 37.51 (1.43×) | 43.59 (1.66×) | 82.03 (3.12×) |

## B.3. Ablation Study

We start from the naive 1F1B-2nd baseline and progressively enable SymFold, WaveK, and GARS. SymFold removes redundant recomputation and synchronization. WaveK reduces pipeline bubbles by improving overlap under the four-phase dependency partial order. GARS mitigates micro-batch imbalance and reduces halo synchronization overhead under GP. Figure 11 reports the normalized throughput improvement over 1F1B-2nd. Overall, the three components are complementary. SymFold improves throughput by up to 23% by eliminating redundant recomputation and avoiding cross-device replicated parameter synchronization at optimizer-step boundaries. WaveK further improves throughput by 0–18% under a fixed memory budget by selecting an effective unit size $k$. GARS contributes an additional 6–23% by balancing micro-batches and reducing GP-induced stalls.

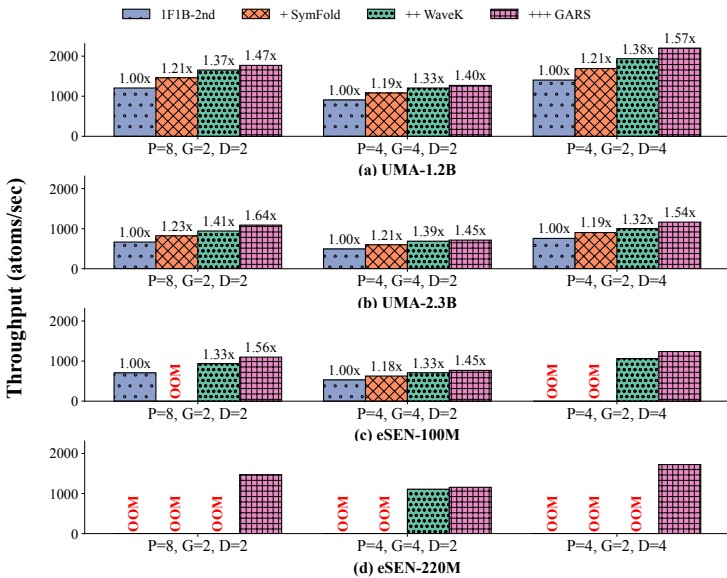

*Figure 11.* Normalized throughput (atoms/sec) over 1F1B-2nd, progressively enabling SymFold, WaveK, and GARS.

## B.4. WaveK Sensitivity Analysis

We sweep candidate values of the WaveK unit size $k$ for JanusPipe on UMA-1.2B with $P=4$, $G=D=1$. Figure 12 shows throughput (normalized to $k=4$) and peak device memory. All runs use JanusPipe+1F1B-2nd, and $k$ is set manually for each evaluation. Increasing $k$ reduces pipeline bubbles and improves throughput up to $k=8$ (peaking at $1.05\times$), while peak memory grows steadily. When $k$ does not evenly divide $N_{\mathrm{mb}}$ (here $N_{\mathrm{mb}} = 32$), the last wave is not fully filled, creating a trailing-wave bubble and causing throughput to drop at $k=5$ and $k=9$. At $k=11$, peak memory reaches the 40.96 GB device limit and the run fails with OOM. The best trade-off is achieved at $k=8$, which maximizes throughput without exceeding the memory bound. We evaluate a few candidate $k$ values for five iterations each and select the best, performing this procedure only once per training run. By default, $k=P$ incurs no search overhead and already yields competitive throughput with minimal memory. Overall, throughput is relatively insensitive near the optimum: multiple neighboring $k$ values deliver similar performance, while the main drops come from non-divisible $k$ (trailing-wave bubbles) or exceeding the memory bound. Therefore, a coarse sweep over a small candidate set (or the default $k=P$) is sufficient in practice.

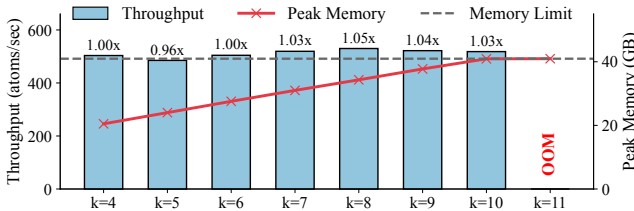

*Figure 12.* UMA-1.2B ($P{=}4$, $G{=}D{=}1$): throughput and peak memory under varying wave size $k$.

## B.5. Bubble Analysis

To evaluate scheduling efficiency, we analyze the pipeline bubble ratio on UMA-1.2B with $P = 4$ and $N_{mb} = 12$ using profiler traces. As shown in Figure 13, we compare the execution timelines under different WaveK unit sizes ($k$). For $k = 6$, the measured bubble time (GPU idle time due to four-phase dependencies and imbalance) is 1295.92 ms, accounting for 21.23% of the total step time (6105.23 ms). Increasing $k$ to 12 reduces the bubble time to 1186.06 ms (19.89%), improving GPU utilization to 80.11%.

As discussed in Section 4.2, the boundary-induced bubble term decreases approximately inversely with $k$. In practice, the observed bubble ratio also includes *imbalance-induced* stalls: residual load imbalance across micro-batches can shift bubbles along the pipeline and propagate to downstream stages. Therefore, while a larger $k$ effectively amortizes boundary bubbles, its end-to-end speedup can be smaller than the ideal prediction when imbalance-induced bubbles become more pronounced. Additionally, the analysis in Section 4.2 focuses on boundary-induced bubbles, while warm-up and cool-down bubbles introduce additional overhead in practice.

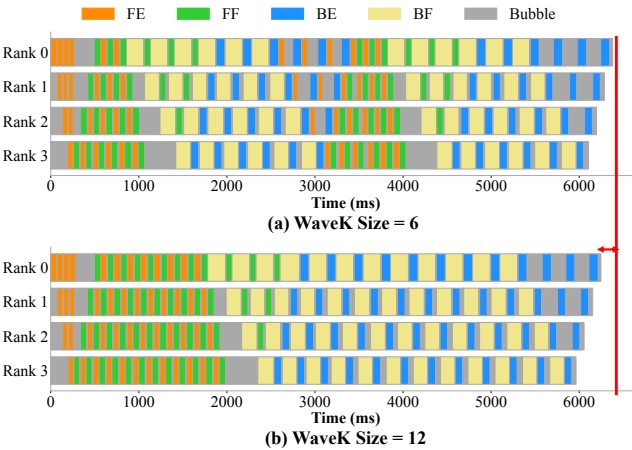

*Figure 13.* Pipeline execution timelines on UMA-1.2B ($P = 4$).

## B.6. GARS Micro-benchmarks

We micro-benchmark the impact of GARS on communication and load balance. We use UMA-1.2B and compare GARS against the same schedule without repacking, under identical global batch and parallelism settings. Figure 14 reports throughput and halo All-Gather time. Without shuffling, graphs within each micro-batch are sorted from large to small. This ordering amplifies GP imbalance, because edge counts grow superlinearly with atom count and large graphs tend to dominate a few ranks. As a result, throughput drops (e.g., about 7% at $P{=}4, G{=}4, D{=}2$). With a single shuffle per micro-batch, throughput improves across all $G{>}1$ settings (up to 14% at $P{=}4, G{=}2, D{=}4$). The halo All-Gather time also drops substantially (e.g., from 70.92 ms to 43.88 ms at $P{=}4, G{=}2, D{=}4$).

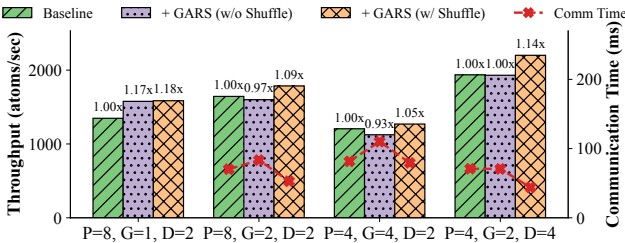

*Figure 14.* UMA-1.2B: throughput (left y-axis) and halo All-Gather time (right y-axis) with SymFold+WaveK.

**GARS mitigates micro-batch imbalance.** Across 1,000 iterations, GARS maintains a consistently low standard deviation of per-micro-batch atom counts (Figure 15), indicating more balanced packing across micro-batches and, consequently, reduced straggler effects across pipeline stages and DP ranks. The baseline uses naive fixed-atom packing with greedy sequential construction of micro-batches, without repacking or shuffling.

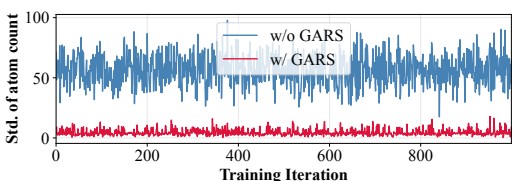

*Figure 15.* Per-iteration standard deviation of micro-batch atom counts over 1,000 iterations, with and without GARS. Lower values indicate more balanced micro-batches.

## B.7. Scalability Analysis

We report strong and weak scaling results on UMA-2.3B. In strong scaling, we fix the total problem size and increase the number of devices. In weak scaling, we proportionally increase the global batch size with the number of devices. Figure 16 summarizes both results. JanusPipe achieves 70% strong-scaling efficiency and 94% weak-scaling efficiency, and delivers up to $1.50\times$ higher throughput than 1F1B-2nd at 32 devices.

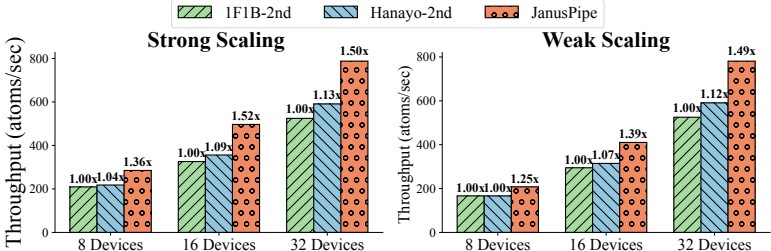

*Figure 16.* Scalability analysis: strong scaling (left) and weak scaling (right).

## B.8. Correctness Validation

**Gradient Computation Correctness.** Equation 2 shows that our gradient merging preserves the mathematical correctness of parameter updates. In non-pipelined training, the total gradient $\frac{\partial L_{\text{total}}}{\partial \theta}$ naturally combines contributions from both energy and force losses. As shown in Equation 1, the parameter gradients decompose into three terms. BE backpropagates through FE and contributes the first-order term. Because FF is obtained by differentiating FE, BF must backpropagate through the FE graph as well as the FF graph. Consequently, BF includes both a first-order term and a second-order term, and we merge the BF first-order term into BE in our implementation. Specifically, BE accumulates all first-order gradient contributions (from both $L_E$ and $L_F$), while BF handles only the second-order term arising from $\frac{\partial L_F}{\partial (\frac{\partial E}{\partial h_i})} \cdot \frac{\partial^2 E}{\partial h_i \partial \theta}$. Since the parameter update depends only on the total gradient $\frac{\partial L_{\text{total}}}{\partial \theta}$, and our transformation preserves this gradient, enabling pipeline parallelism

does not change the mathematical update rule compared to the non-pipelined baseline ($P=1$).

**Empirical Verification.** We validate this correctness guarantee by comparing the energy/force MAE trajectories of JanusPipe against a no-PP reference run under identical training conditions: the same model and dataset, identical optimizer hyperparameters, the same parameter initialization, and fixed random seeds. The only difference is whether pipeline parallelism is enabled ($P=4$ vs. $P=1$), while GP/DP settings remain the same. We enable deterministic PyTorch execution whenever possible (e.g., deterministic algorithm settings and deterministic cuBLAS/cuDNN behavior). To avoid OOM in the reference run, we reduce the micro-batch size while keeping the global batch size unchanged via gradient accumulation.

Figure 17 plots MAE trajectories over 1,000 training iterations. The trajectories closely match, with mean absolute percentage errors of 0.84% for energy MAE and 0.21% for force MAE. The small residual discrepancies are attributable to non-associativity in floating-point arithmetic under distributed execution (e.g., different reduction/aggregation orders across pipeline stages), which is expected; empirically, both runs exhibit similar convergence behavior.

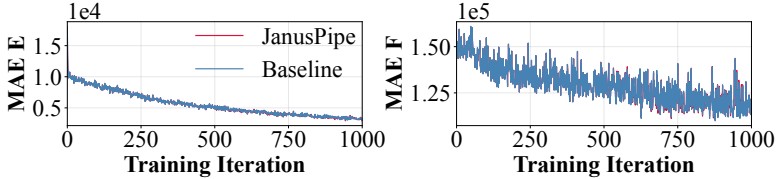

*Figure 17.* Correctness validation on UMA-1.2B. We compare JanusPipe ($P=4, G=4, D=2$) against a no-PP reference ($P=1, G=4, D=2$) for 1,000 training iterations. The nearly overlapping curves indicate that enabling PP does not change the training trajectory.

