# OpenReview forum: "JanusPipe: Efficient Pipeline Parallel Training for Machine Learning Interatomic Potentials"
_ICML.cc/2026/Conference — ICML 2026 regular_

### Official Review · Reviewer_4KpU · 2026-02-25

**Soundness:** 3
**Presentation:** 4
**Significance:** 3
**Originality:** 4
**Overall Recommendation:** 5
**Confidence:** 4

**Summary:**

This paper proposes JanusPipe, an efficient parallel training pipeline specifically designed for conservative machine learning interatomic potentials (MLIPs). The authors point out that conservative  MLIPs are second-order models (requiring gradient computation to obtain forces), and existing first-order pipeline schedules suffer from issues such as redundant computation and extra memory usage. Based on the characteristics of MLIP models (such as the consistent partial order in the execution times of their four phases), new scheduling components, including SymFold, WaveK, and GARS, were designed. Experimental results show that JanusPipe achieved notable improvements in throughputs and memory consumptions.

**Compliance With Llm Reviewing Policy:**

Affirmed.

**Final Justification:**

All my concerns are addressed in the rebuttal. I recommend acceptance of this paper and it will be valuable to the field.

**Key Questions For Authors:**

See weaknesses.

**Limitations:**

yes

**Strengths And Weaknesses:**

Strengths:
- This work clearly identifies and analyzes the fundamental mismatch problem of parallel training of conservative MLIPs. This is a bottleneck in large-scale training of MLIPs.
- The authors propses a systematic solution and the experimental validation is comprehensive. The tested models include both the sparse UMA and the dense eSEN, and different settings (PP/GP/DP) are evaluated, covering key metrics such as throughput and memory efficiency.

Weaknesses:
- JanusPipe increases the implementation complexity and debugging difficulty of the system, which may pose a barrier for general researchers and engineers. How to apply JanusPipe to one's own models is not discussed.
- The paper primarily focuses on improving training efficiency (throughput, memory). Although correctness validation is provided in the appendix, it does not present the final convergence accuracy. Whether such complex scheduling modifications would have impact on the final convergence accuracy and training stability, particularly under extremely large-scale training or different hyperparameters? This is a point that requires discussion.

---

> ### Author Rebuttal · Authors · 2026-03-31
>
> We sincerely thank the reviewer for the encouraging feedback, especially for recognizing that our work **clearly identifies the fundamental mismatch in parallel training of conservative MLIPs and proposes a systematic solution with comprehensive experimental validation.** We also thank the reviewer for raising two important practical questions: how easily JanusPipe can be integrated into researchers’ own models, and whether it affects convergence accuracy or training stability.
>
>
>
> > Q1: How to apply JanusPipe in practice
>
> We agree that usability and ease of integration are important practical concerns for a systems contribution.
>
> Specifically, using JanusPipe does **not** require:
>
> - hand-writing a four-phase (FE/FF/BF/BE) pipeline schedule;
>
> - modifying the backbone internals of the MLIP model (e.g., GNN layers).
>
>
> Instead, the main integration effort is a one-time integration adapter, which:
>
> - wraps the model into stage-partitioned modules, and
>
> - maps the original training step to the JanusPipe execution interface.
>
>
> Therefore, the **user-facing training loop change is minimal**, and users should just replace the original `train_step` with `engine.execute_step()`.
>
> ```Python
> # before
> model = Model(...)
> trainer = Trainer(model=model, optimizer=optimizer, ...)
> for batch in dataloader:
>     trainer.train_step(batch)
>
> # after
> model = janus.init_stage_model(Model, partition=..., ...)
> trainer = Trainer(model=model, optimizer=optimizer, ...)
> engine = JanusEngine.from_trainer(trainer, ...)
>
> for batch in dataloader:
>     engine.execute_step(batch)
> ```
>
> This design keeps most scheduling and execution complexity inside JanusPipe, allowing researchers to apply pipeline parallelism without manually handling the four-phase dependencies.
>
> In addition, we have further applied JanusPipe to PET-1.2B and MatRIS-M models, where it improves throughput by $1.42\times$–$1.67\times$ over 1F1B. These results further support the generality of JanusPipe across conservative MLIP backbones that follow the same four-phase training workflow.
>
> We will add a dedicated usage subsection in the final version and plan to open-source the code with documentation and a minimal integration example to make this integration path clearer and easier.
>
> > Q2: Convergence and training stability
>
> In Appendix B.8 and Figure 17, the correctness validation shows that JanusPipe preserves the training trajectory over the first 1,000 iterations.
>
> **During the rebuttal period, we have added a longer MatRIS-M training experiment of 40,000 iterations under the same hyperparameters.** Specifically, we used a fixed mixture of OC, OMol25, OMat24, and OMC. We subsampled 1k samples from each dataset, used a sampling ratio of [1.0, 2.0, 4.0, 1.0], and compared DDP4 with two PP4-GP4 runs using different seeds for up to 40,000 iterations.
>
> The resulting metrics remain highly consistent, and the final accuracies at 40k iterations are:
>
> |   |   |   |   |   |
> |---|---|---|---|---|
> |Iter.|Metric|DDP4|PP4-GP4-s2025|PP4-GP4-s2026|
> |4k|MAE-E|0.1014|0.0981|0.0997|
> |4k|MAE-F|0.1908|0.1963|0.1962|
> |4k|F-cos|0.5380|0.5286|0.5295|
> |20k|MAE-E|0.0541|0.0522|0.0538|
> |20k|MAE-F|0.1053|0.1033|0.1045|
> |20k|F-cos|0.6758|0.6777|0.6803|
> |40k|MAE-E|0.0362|0.0361|0.0362|
> |40k|MAE-F|0.0722|0.0726|0.0721|
> |40k|F-cos|0.7539|0.7540|0.7610|
>
> The metrics are already very close from the middle stage of training onward (e.g., 15,000 to 20,000 iterations), and we do not observe degraded final accuracy or training instability caused by JanusPipe.
>
> This is expected because JanusPipe improves distributed MLIP training efficiency and scalability **without changing the training objective or numerical behavior**. It is based on the following guarantees:
> - Correct gradient update path. Its IR design, schedule transformation, and mathematical formulation preserve the original double-backward gradient update path.
> - Unchanged update granularity. JanusPipe only rebalances samples within a global batch, without changing the training objective.
>
> > **Summary**
>
> We thank the reviewer again for the valuable feedback. In response to the reviewer’s practical questions, we have clarified how JanusPipe can be integrated into new conservative MLIP backbones, and we have added longer-run evaluation on training stability. We will include these results and the corresponding discussion in the final revision.
>
> We will also add a dedicated usage subsection in the final version. **In addition, we plan to open-source the code with documentation and a minimal integration example to make this process clearer and easier for researchers.** We recognize that distributed training adds complexity, especially when scaling conservative MLIPs. Still, **we will do our best to reduce this burden by improving usability, providing better documentation, and supporting broader adoption in the MLIPs community. We hope these efforts will lower the barrier to adopting JanusPipe and help future scaling-law studies in MLIPs.**

---

> > ### Author Rebuttal · Reviewer_4KpU · 2026-04-01
> >
> > Thank you for your response. All my questions have been resolved, and I will increase my score. I look forward to using JanusPipe.

---

> > > ### Author Response · Authors · 2026-04-01
> > >
> > > We sincerely thank the reviewer for the positive feedback and for carefully considering our responses. We also appreciate the reviewer’s interest in JanusPipe.

---

### Official Review · Reviewer_ZPX8 · 2026-03-12

**Soundness:** 3
**Presentation:** 3
**Significance:** 3
**Originality:** 3
**Overall Recommendation:** 4
**Confidence:** 4

**Summary:**

This paper presents a pipeline parallelism scheduling for Machine Learning Interatomic Potentials (MLIP). It is presented as a sequence of transformations from the classical 1F1B scheduling, categorized as SymFold and WaveK. It achieves 1.51x speedup over 1F1B on 32 GPUs.

**Compliance With Llm Reviewing Policy:**

Affirmed.

**Final Justification:**

Thanks authors for the clear answers of my concerns. It is a technically solid paper that advances at least one sub-area of AI (MLIPs). The paper is well written and presented, easy to follow. The elegant transformation formulation and the parametric number of living micro batch could be inspirational for pipeline scheduling future works. However, it is still with some weaknesses that limit its impact: for a PP paper, the number of GPUs used in exp is small; and Offline Tuning under Memory Constraints is critical for PP performance, but the profiling-based method may be expensive for cases on a big cluster.

**Key Questions For Authors:**

For key questions please see weakness points above.

**Limitations:**

No, the limitations and potential societal impact discussion is not developed.

**Strengths And Weaknesses:**

Strengths:
- The paper is well written and presented, easy to follow.
- The paper makes pipeline parallelism a much better option for MLIP, thereby improving its scaling
- The elegant transformation formulation and the parametric number of living micro batch could be inspirational for pipeline scheduling future works
- Evaluation includes multiple model families and 3D parallel configurations, with ablations and scaling studies

Weaknesses:
- Baseline fairness is a concern: adapted 1F1B/Hanayo baselines are penalized by framework constraints (e.g., difficulty transferring autograd graph state/activations across processes), which forces FE recomputation on the FF side. A stronger baseline that enforces FE/FF co-location (even a naive folding without WaveK) would better isolate incremental benefits and reduce perceived asymmetry.
-  There are some concerns on the math for the Bubble Analysis in section 4.2, from l.271. :
    - The intra-unit bubble size is written to be tBE – tFF but from Figure 5, it looks to be tBF – tFE (which is incidentally the same as the inter-unit). If it is really tBE – tFF then it needs to be explained.
    - The inter-unit bubble factor is written to be the number of waves (MB / k); however, it only appears between the waves, so it should be MB/k -1. It cannot extend to the global pipeline warm-up (P*tFF) or cool-down (P*BF) because it is not equal.
- some scheduling concepts (wave-style overlap, bubble reduction) have precedented in existing PP literature; the novelty relies on convincingly arguing that existing programmable PP systems cannot be straightforwardly extended to the four-phase second-order setting and on empirical demonstration of that gap.
- Overheads of instruction-list generation/execution and GARS runtime costs are described mainly qualitatively; more quantitative breakdowns would improve clarity.
- Evidence of robustness/external validity is limited: results are capped at 32 GPUs and one hardware class; broader scaling (multi-node, 64–128 GPUs) and cross-platform validation are missing.

---

> ### Author Rebuttal · Authors · 2026-03-31
>
> We sincerely thank the reviewer for the valuable and constructive feedback, and especially for highlighting several strengths of our work: **a well-written and easy-to-follow presentation, an elegant transformation formulation that may inspire future work, a more effective option for scaling MLIPs, and a comprehensive evaluation.**
>
> > Q1: Baseline fairness and a stronger folded baseline
>
> To address this concern, we additionally report the performance of 1F1B-2nd+SymFold, a stronger folded baseline that enforces FE/FF co-location but excludes WaveK and GARS.
>
> |   |   |   |   |
> |---|---|---|---|
> |Config (32 A100 GPUs)|1F1B-2nd|1F1B-2nd + SymFold|JanusPipe|
> |PP4-GP4-DP2|1.00×|1.21×|1.45×|
> |PP4-GP2-DP4|1.00×|1.19×|1.54×|
> |PP8-GP2-DP2|1.00×|1.23×|1.64×|
>
> Our paper already includes this ablation in Section 5.3 and Appendix B.3. Consistent with that ablation, JanusPipe still outperforms this stronger folded baseline on all tested 32-GPU settings, indicating that the gains do not come from FE/FF co-location alone.
>
> > Q2: Bubble analysis in Section 4.2
>
> We appreciate the reviewer’s careful proofreading. We agree that Section 4.2 was not stated precisely enough and will revise it.
>
> 1. For the first point, the intended intra-unit residual is $t_{BE}-t_{FF}$. Within a WaveK unit, Pass 5 inserts the residual FF work from the end of WaveK-F into the initial BE-dominated bubble at the start of WaveK-B. Figure 5 shows the inter-unit boundary bubble more clearly, whose visible residual is $t_{BF}-t_{FE}$; the intra-unit residual is less visible because the schematic sets $t_{BE}$ approximately equal to $t_{FF}$.
>
> 2. For the second point, we agree that the inter-unit count should be $\frac{Nmb}{k} - 1$ rather than $\frac{Nmb}{k}$ . We also agree that the simplified counting in Section 4.2 did not separately include warm-up and cool-down overhead, which should be stated explicitly.
>
>
> We will revise Section 4.2 accordingly by explicitly separating the intra-unit, inter-unit, and warm-up/cool-down terms and by aligning the description more carefully with Figure 5. This is a clarification of the analysis only and does not affect the schedule itself, the paper’s main conclusions, or the evaluation results.
>
> > Q3: Relation to prior PP literature and novelty boundary
>
> We agree that the scheduling concept itself is not new. Our claim is more specific: existing PP schedules assume a two-phase FW/BW dependency, whereas conservative MLIPs with double backward require a four-phase FE/FF/BF/BE execution pattern with different dependency constraints and memory demands. **To our knowledge, no prior work directly addresses this four-phase double backward workload, whereas JanusPipe is designed specifically for this challenge.**
>
> The main contributions of JanusPipe are: a device-independent instruction-list abstraction of conservative MLIP training, together with automatic schedule transformations that (i) co-locate FE and FF, (ii) safely increase overlap across the four phases, and (iii) mitigate micro-batch imbalance.
>
> > Q4: Quantitative overheads of instruction generation and GARS runtime
>
> We agree that these overheads should be quantified. We now report both the instruction generation cost and the per-iteration GARS runtime overhead (10 warm-up iterations, 100 repeated iterations):
>
> |   |   |   |   |
> |---|---|---|---|
> |Model config|GARS runtime cost|Instruction generation cost|Total overhead ratio|
> |PP4-GP4-MB16|1.78 ms|0.30 ms|0.028%|
> |PP8-GP2-MB16|1.78 ms|0.30 ms|0.031%|
> |PP4-GP4-MB32|4.06 ms|0.30 ms|0.069%|
> |PP8-GP2-MB32|4.07 ms|0.30 ms|0.064%|
>
> The combined overhead of instruction list generation and GARS remains below 0.1% of the end-to-end iteration time in all tested settings.
>
> > Q5: External validity beyond 32 GPUs and one hardware class
>
> We agree that broader validation is valuable. To strengthen this point, we have added experimental results on both a larger-scale platform (i.e., 64 A100-PCIe GPUs) and a new NVLink platform (i.e., 32 A800 NVLink GPUs on 4 nodes with cross-node IB connections). On both platforms, JanusPipe shows the same improvement trends.
>
> |   |   |   |   |
> |---|---|---|---|
> |Config (64 A100-PCIe GPUs)|1F1B-2nd|1F1B-2nd + SymFold|JanusPipe|
> |PP4-GP4-DP4|1.00×|1.23×|1.52×|
> |PP4-GP2-DP8|1.00×|1.11×|1.42×|
> |PP8-GP2-DP4|1.00×|1.25×|1.68×|
>
> |   |   |   |   |
> |---|---|---|---|
> |Config (32 A800-NVLink GPUs)|1F1B-2nd|1F1B-2nd + SymFold|JanusPipe|
> |PP4-GP4-DP2|1.00×|1.17×|1.44×|
> |PP4-GP2-DP4|1.00×|1.16×|1.52×|
> |PP8-GP2-DP2|1.00×|1.22×|1.62×|
>
> > Summary
>
> We again thank the reviewer for the careful reading and constructive suggestions. We will incorporate the additional evaluations and discussion described above in the final revision. Specifically, we will revise the bubble analysis in Section 4.2, add a quantitative overhead analysis, and include the 64-GPU and NVLink evaluations in the evaluation section.

---

> > ### Author Rebuttal · Reviewer_ZPX8 · 2026-04-02
> >
> > Author answered most of myquestions. For my concern on the prior PP literatur, the four-phase double backward workload is very specific with limited applications which reduce the value of the contribution. Meanwhile, this paper is NOT the first to study multi-phase PP workloads, we also use multi-phase PP workloads on Interleaving PP, Multimodal PP, etc.

---

> > > ### Author Response · Authors · 2026-04-02
> > >
> > > We thank the reviewer for the helpful follow up comments on the novelty and scope of our work.
> > >
> > > > Q1: The double-backward workload is very specific and has limited applications.
> > >
> > > We agree that this workload is specific to conservative MLIPs. However, we believe it is still important. Molecular dynamics simulation is widely used in materials science, chemistry, and related areas. Conservative MLIPs are important because they enable fast and accurate simulation, while also supporting stable long-running MD simulation. This makes scalable training for conservative MLIPs an important systems problem.
> > >
> > > > Q2: Comparison with prior multi-phase PP workloads.
> > >
> > > We agree that prior work has explored multi-phase PP workloads, but this is not the same problem. JanusPipe is a PP design tailored to conservative MLIP training with a four-phase double-backward workflow.
> > >
> > > In multimodal PP systems such as DIP [1], DISTMM [5], and Optimus [6], the multi-phase execution comes from the model having different components for different modalities. The main challenge is dynamic imbalance caused by heterogeneous modules and varying input composition. In contrast, our scenario is quite different: the core challenge comes from the four-phase **double-backward gradient dependencies** in conservative MLIPs, including FE activation reuse in FF and correct higher-order gradient paths across BF/BE.
> > >
> > > Other prior PP work also studies related but different scenarios. Interleaved PP [3] improves first-order forward/backward utilization with virtual stages. Bidirectional / wave-style PP [2,4] reduces bubbles for standard first-order training. These workloads do not need to preserve the same double-backward dependencies as conservative MLIPs.
> > >
> > > > Summary
> > >
> > > Thank you again for this helpful comment. We hope this work can provide a useful contribution to the MLIP community. In the final version, we will expand the related work section and add citations to prior multi-phase pipeline parallelism work. This will help clarify how JanusPipe differs from these systems.
> > >
> > > [1] Zhenliang Xue et al., DIP: Efficient Large Multimodal Model Training with Dynamic Interleaved Pipeline, ASPLOS 2026.
> > >
> > > [2] Shigang Li et al., Chimera: Efficiently Training Large-Scale Neural Networks with Bidirectional Pipelines, SC 2021.
> > >
> > > [3] Deepak Narayanan et al., Efficient Large-Scale Language Model Training on GPU Clusters Using Megatron-LM, SC 2021.
> > >
> > > [4] Ziming Liu et al., Hanayo: Harnessing Wave-like Pipeline Parallelism for Enhanced Large Model Training Efficiency, SC 2023.
> > >
> > > [5] Jun Huang et al., DISTMM: Accelerating Distributed Multimodal Model Training, NSDI 2024.
> > >
> > > [6] Weiqi Feng et al., Optimus: Accelerating Large-Scale Multi-Modal LLM Training by Bubble Exploitation, USENIX ATC 2025.
> > >
> > > **Response to final justification weaknesses**
> > >
> > > ---
> > >
> > > We thank the reviewer for the positive comments.
> > > > Q1: The number of GPUs used is small
> > >
> > > We added 64-GPU experiments in ``Rebuttal Q5`` to address this concern.
> > > Our main scheduling optimization targets PP. In a larger cluster, adding more GPUs is usually done by increasing DP. Since DP communication overlaps with backward, it does not change the core issue we study, namely the double-backward PP scheduling behavior. Therefore, the 64-GPU results are sufficient to support the big cluster behavior of our method.
> > >
> > > > Q2: The WaveK offline tuning method is critical for PP performance and expensive.
> > >
> > > **First, tuning is not critical for PP performance.**
> > > ``Section 5.2`` shows that default `k=P` is already competitive, with mean speedup improving from 1.45× to 1.50× after tuning.
> > >
> > > **Second, the tuning overhead is small compared with the training cost.**
> > > This is already described in ``Section 4.2``. WaveK selects `k` by a one-time offline search before training. If tuning is skipped, the default `k=P` can be used. Each candidate `k` only needs a few profiling steps.
> > > We also report the overhead under PP=4:
> > >
> > > |                |          |          |
> > > | -------------- | -------: | -------: |
> > > | overhead (min) | UMA-1.2B | UMA-2.3B |
> > > | total time     |    5.958 |   14.675 |
> > >
> > >
> > > Thus, tuning is lightweight.
> > >
> > > **Finally, it does not require full-cluster tuning.**
> > > WaveK tunes an intra-PP-group parameter `k`. In the paper, profiling does not include DP groups; a PP group is sufficient. We further compare PP=4, DP=1 and PP=4, DP=8. The optimal or near-optimal `k` is the same in both settings (around `k=8`):
> > >
> > >
> > > |        |            |            |
> > > | ------ | ---------: | ---------: |
> > > | k size | PP=4, DP=1 | PP=4, DP=8 |
> > > | 4      |      1.00× |      1.00× |
> > > | 5      |      0.96× |      0.95× |
> > > | 6      |      1.00× |      1.01× |
> > > | 7      |      1.03× |      1.04× |
> > > | 8      |      1.05× |      1.06× |
> > > | 9      |      1.04× |      1.04× |
> > > | 10     |      1.03× |      1.02× |
> > >
> > >
> > > This matches the sensitivity analysis in ``Appendix B.4``. This shows that when scaling mainly increases DP, the choice of `k` stays stable. Therefore, full-cluster tuning is unnecessary.

---

### Official Review · Reviewer_XFZf · 2026-03-13

**Soundness:** 3
**Presentation:** 3
**Significance:** 3
**Originality:** 3
**Overall Recommendation:** 5
**Confidence:** 3

**Summary:**

This paper proposed a new training pipeline for MLIPs. They first identified bubbles from the four phase of MLIP training. Then proposed 3 methods to get better parallelism: 1. SymFold: A schedule transformation that co-locates FE and FF on the same device via symmetric folding of 2P virtual stages onto P physical devices. 2. WaveK: Groups k micro-batches into scheduling units, with forward and backward waves that overlap at unit boundaries to reduce bubbles. 3. GARS: A lightweight graph-aware micro-batch repacking heuristic to handle the long-tailed graph size distributions. Results showed 1.5x speed up on 32 A100s.

**Compliance With Llm Reviewing Policy:**

Affirmed.

**Final Justification:**

The rebuttal has addressed my concerns, and I think the approach is a valuable contribution to the field. Thus I would recommend this paper for acceptance.

**Key Questions For Authors:**

1. Did you test with other models?
2. Did you test with other hardware? Especially NVLink / DGX?

**Limitations:**

Yes

**Strengths And Weaknesses:**

Strengths
1. This is a well-motivated problem. There are no prior work (that I am aware of) that exploit the system-level bottleneck of MLIP training. The double backward training is unique to MLIP training and has been very inefficient
2. The IR-based approach (passes 0–3 for SymFold, passes 4–6 for WaveK) is elegant. It makes the schedule transformations composable and easier to reason about for correctness. The separation into well-defined passes is a good systems design choice.
3. The experiments are comprehensive, including compute/memory profiles, multiple parallelism configurations, and ablations, sensitivity analysis, bubble breakdowns, correctness validation, micro-benchmarks.

Weakness
1. The approach is only evaluated on one type of model (UMA and eSEN is the same design). It's unclear about the generalizability of the method.
2. The authors only test on one type of hardware, A100-40GB PCIe. The PCIe interconnect is much slower than NVLink, which could significantly affect the PP communication costs and relative benefits. It's unclear how results would transfer to NVLink-connected clusters where communication overhead is lower and the bubble/communication balance shifts.

---

> ### Author Rebuttal · Authors · 2026-03-30
>
> We sincerely thank the reviewer for the valuable and constructive feedback, and especially for pointing out three key strengths of our work: **a well-motivated systems problem in MLIP training, an elegant IR-based design, and a comprehensive evaluation.**
>
> Inspired by your suggestions, we have **1) added experiments on two additional conservative MLIP models, PET [1] and MatRIS [2]**, and also **2) evaluated JanusPipe on an** **NVLink** **platform**. These results can strengthen both the generality and portability of JanusPipe. We will incorporate them and the corresponding analysis in the final revision.
>
> > Q1: Evaluation of other models
>
> **We have added evaluations on PET-1.2B [1] and MatRIS-M [2] during the rebuttal period.** We report the results in the table below. JanusPipe achieves consistent throughput improvements over 1F1B-2nd and Hanayo-2nd.
>
> |   |   |   |   |   |
> |---|---|---|---|---|
> |MLIP Model|Parallel Config|1F1B-2nd|Hanayo-2nd|JanusPipe|
> |PET-1.2B|PP4-GP4-DP2|1.00×|1.05×|1.44×|
> ||PP8-GP2-DP2|1.00×|1.14×|1.62×|
> ||PP4-GP2-DP4|1.00×|1.07×|1.52×|
> |MatRIS-M|PP4-GP4-DP2|1.00×|1.07×|1.42×|
> ||PP8-GP2-DP2|1.00×|1.08×|1.67×|
> ||PP4-GP2-DP4|1.00×|1.04×|1.57×|
>
> These results show that JanusPipe generalizes beyond UMA/eSEN. This is because JanusPipe operates at the distributed runtime/schedule level rather than being specialized to a particular MLIP backbone. Specifically, it abstracts the common four-phase conservative training workflow (FE/FF/BF/BE), instead of relying on model-specific layer implementations or hand-written schedules for each backbone.
>
> We will add a short usage subsection and a minimal code example in the final version to make this integration path clearer. We also plan to open-source the code with documentation to make JanusPipe easier to adopt for other conservative MLIP backbones.
>
> > Q2: Evaluations on other hardware, especially NVLink / DGX
>
> **We have added experiments for UMA-2.3B on an A800** **NVLink** **platform during the rebuttal period, to clarify whether JanusPipe’s improvement depends on the slower** **PCIe** **interconnect.**
>
> |   |   |   |   |   |
> |---|---|---|---|---|
> |Platform|Config|PP comm ratio / step|PP bubbles (GPU idle time) / step|JanusPipe's speedup over 1F1B-2nd|
> |A100-PCIe|PP4-GP4-DP2|1.72% (1.14 GB)|19.73%|1.45×|
> |A800-NVLink|PP4-GP4-DP2|0.62% (1.14 GB)|19.21%|1.44×|
> |A100-PCIe|PP4-GP2-DP4|1.91% (2.29 GB)|16.39%|1.54×|
> |A800-NVLink|PP4-GP2-DP4|0.92% (2.29 GB)|16.25%|1.52×|
>
> Experimental results show that JanusPipe achieves similar end-to-end speedups across NVLink and PCIe platforms. For UMA-2.3B on PP4-GP4-DP2, the speedup over 1F1B-2nd is 1.44× on NVLink and 1.45× on PCIe. On PP4-GP2-DP4, it is 1.52× on NVLink and 1.54× on PCIe.
>
> To explain this, we categorize the pipeline parallel (PP) overhead into two parts: 1) PP communication and 2) PP bubbles (i.e., GPU idle time). PP communication is already a small fraction of the step time. As shown in the above table, under PP4-GP4-DP2, it is 1.72% on PCIe and 0.62% on NVLink, which suggests that device-to-device bandwidth is not the main bottleneck. The largest overhead comes from PP bubbles, caused by FE/FF/BF/BE instruction dependencies and inter-microbatch load imbalance. JanusPipe therefore focuses on reducing PP bubbles through SymFold, WaveK, and GARS optimizations.
>
> As expected, NVLink reduces the communication ratio, but the overall speedup changes quite slightly. This means JanusPipe does not rely on slower PCIe links. We have no DGX platform, but the NVLink results can effectively address the concern about interconnect dependence.
>
> > Summary
>
> We thank the reviewer again for raising these questions about model generality and evaluation on an NVLink platform. We will add the PET/MatRIS results and the NVLink analysis to the Evaluation section. These revisions will make the paper more solid and support our claim: JanusPipe is effective not only on UMA/eSEN, but also on other conservative MLIP backbones, and it remains effective on both PCIe and NVLink platforms.
>
> [1] Filippo Bigi et al., Pushing the Limits of Unconstrained Machine-Learned Interatomic Potentials, arXiv 2026.
>
> [2] Yuanchang Zhou et al., MatRIS: Toward Reliable and Efficient Pretrained Machine Learning Interatomic Potentials, ICLR 2026.

---

> > ### Author Rebuttal · Reviewer_XFZf · 2026-04-01
> >
> > Thanks the authors for the reply. I have a few clarification problems:
> >
> > 1. What is the specification of the A800-NVLink platform? How many GPUs, and what's the connection topology of GPUs? (ie how is the NVLink configured, group of 2, 4, or 8?)
> > 2. For UMA-2.3B, is it a dense model or MoE version? Did you use FSDP over experts?
> >
> > Edit: Thanks the authors again for the quick reply, I am satisfied and raised my score

---

> > > ### Author Response · Authors · 2026-04-01
> > >
> > > We thank the reviewer for the follow-up questions.
> > >
> > > > Q1: What is the specification of the A800-NVLink platform? How many GPUs, and what's the connection topology of GPUs? (ie how is the NVLink configured, group of 2, 4, or 8?)
> > >
> > > The A800-NVLink platform is the **HGX A800-8GPU platform**, which consists of 8 NVIDIA A800 GPUs in a single node and cross-node communication uses InfiniBand (IB). According to `nvidia-smi topo -m`, every pair of GPUs is connected via NV8; that is, each GPU has direct NVLink connectivity to the other 7 GPUs. Therefore, the intra-node topology is a fully connected 8-GPU NVLink topology.
> > >
> > > > Q2: For UMA-2.3B, is it a dense model or MoE version? Did you use FSDP over experts?
> > >
> > > UMA-2.3B is the sparse MoLE-based UMA variant rather than a fully dense model. In our configuration, it uses 128 MoLE experts; however, its computation remains dense because the expert weights are combined by weighted averaging into a temporary linear transform. Specifically, UMA uses a Mixture of Linear Experts (MoLE) rather than a standard MoE formulation, where
> > >
> > > $y = \sum_k \alpha_k (W_k x)$.
> > >
> > > Since the experts are linear, this can be rewritten as
> > >
> > > $y = W^\*x, W^{\*} = \sum_k \alpha_k W_k$.
> > >
> > > Therefore, the computation remains dense after expert-weight mixing.
> > >
> > > We use DP rather than FSDP in this paper. Moreover, combining parameter sharding with the four-phase double-backward execution of conservative MLIPs requires additional system support and is beyond the scope of the current paper (pipeline parallel training).
> > >
> > > In addition, we have conducted preliminary explorations of training standard MLIPs with standard MoE using FSDP+FSEP (Fully Sharded Expert Parallel). As these results are beyond the scope of the current paper, we leave them to future work.
> > >
> > > We thank the reviewer again for the careful reading and thoughtful questions.

---

### Official Review · Reviewer_MEbs · 2026-03-23

**Soundness:** 3
**Presentation:** 2
**Significance:** 2
**Originality:** 3
**Overall Recommendation:** 4
**Confidence:** 4

**Summary:**

The authors develop a parallel training pipeline for conservative MLIPs. This is meant to split up the double backwards differentiation and parallelize these operations.

**Compliance With Llm Reviewing Policy:**

Affirmed.

**Final Justification:**

Thank you to the authors for answering my questions. I would be very interested to see how this framework will enable better training of MLIPs at scale in the future in a way that enables new state-of-the-art, etc.

**Key Questions For Authors:**

- Can you better quantify the advantage of this (fairly complicated) approach for MLIP downstream applications, compared to training an MLIP in a standard fashion? I appreciate the effort that the authors have put into this work, but it’s not fully clear to me that it’s translating into practical impact into a much faster MLIP for running MD, or more accurate (if able to train with more parameters), etc.

- Can you demonstrate the generality of such an approach to other arbitrary types of MLIPs, besides UMA and eSEN (where UMA is built on the eSEN backbone)? As this is a fast moving field with new, well-performing models, it would be helpful to look at other competitive models on some of the current large-scale datasets (OMat24, OMol25, etc.) and see how smoothly such an approach translates there.

- I may have missed this, but I don’t see any place where there’s accuracy benchmarking: does the accuracy stay the same, but the model is faster to train and run at inference, when using this approach? And/or are there any accuracy gains that are also enabled?

- In Table 4, does this mean that an UMA model with 2.3B parameters was trained (and eSEN with 100M, 200M+ parameters)? Does this improve model performance compared to the prior, pre-trained UMA and eSEN models that were much smaller?

**Limitations:**

Limitations are not described.

**Strengths And Weaknesses:**

Strengths:

- This is an interesting systems-based perspective on training conservative MLIPs. I appreciate the detail that the authors have put into describing the full setting.

- I found the overall description of training the MLIP with four phases to be well-described to follow, and why this is split into four phases.

Weaknesses:

- The baselines are hard to contextualize for this field since they are baselines that are not normally used in MLIP training (i.e., the baselines are distributed systems baselines for first-order training): it would be helpful to have baselines to compare training MLIPs this way vs. not training MLIPs this way.

- Related to the above point, I wonder if the approach described in this paper is overkill, given that the speed gains seem relatively modest. More explicit benchmarking and contextualization would be helpful to understand this.

- Prior work has shown that one does not necessarily need to train MLIPs conservatively the full time, instead they can train with direct forces and just fine-tune at the end with conservative forces: this adds minimal cost to the overall MLIP training. Given this, how useful is such an approach in this setting?

---

> ### Author Rebuttal · Authors · 2026-03-30
>
> We sincerely thank the reviewer for recognizing the systems-based perspective of our work and the clarity of the four-phase MLIP formulation.
>
> We understand that the reviewer’s main concern is the practical impact of JanusPipe. In response, we clarify our contributions to the MLIP community, better contextualize our baseline choice, and strengthen the generality discussion with additional PET [3] and MatRIS [4] experiments.
> > **Weakness**
>
> > W1&2: The baselines are not normally used in MLIP training.
>
> Due to GPU memory limits, in the large-scale conservative MLIP setting studied here, single-device or pure data-parallel (DP) training becomes infeasible. Therefore, pipeline parallelism (PP) is the practical distributed training method, as it partitions the model across devices and enables larger models under per-GPU memory limits.
>
> Existing PP schedules target single-backward training, while conservative MLIPs require double backward with four dependent phases (FE/FF/BF/BE). To the best of our knowledge, no dedicated PP method currently supports this scenario.
>
> Therefore, in our evaluation, we focus on how existing PP schedules behave for conservative MLIPs, and how much JanusPipe improves over them. Beyond the 1.51× / 1.45× throughput gains over 1F1B/Hanayo, JanusPipe also reduces peak memory (e.g., from 38.74 GB to 17.6 GB) and makes larger models trainable (especially models suffering from OOM exceptions).
>
> > W3: Necessity of training MLIPs conservatively
>
> We agree that recent MLIP studies have shown that some models use non-conservative pretraining followed by conservative fine-tuning.
>
> However, this does not remove the need for conservative training, since recent studies show that non-conservative MLIPs can violate physical invariants in downstream simulations [1].
>
> **Moreover, even in this first non-conservative then conservative workflow, the conservative stage still dominates the training resources. For example, Meta [2] reports 1,792 H200 GPU-days for the non-conservative stage and 3,584 for conservative stage to train UMA-M, where the conservative stage accounts for 67% of the resources.**
>
> > **Question**
>
> > Q1&3: Downstream impact and accuracy benchmarking
>
> JanusPipe targets **training** efficiency and scalability, not faster MD inference, and we do not claim direct accuracy gains at fixed scale. Its practical impact is that it makes larger conservative MLIP training feasible under realistic memory limits, rather than only in settings where single-device or pure-DP training is feasible.
>
> To verify that JanusPipe preserves training behavior, we additionally ran a longer MatRIS [4] experiment. We observed highly consistent trajectories up to 40k iterations, with final MAE-E = 0.0362 / 0.0361 / 0.0362, MAE-F = 0.0722 / 0.0726 / 0.0721, and F-cos = 0.7539 / 0.7540 / 0.7610, indicating stable convergence. Due to the character limit, please refer to Reviewer 4KpU Q2 for details.
>
> > Q2: Apply JanusPipe to more MLIPs
>
> We agree that evaluating additional models can provide stronger evidence for generality.
>
> We have validated JanusPipe on PET-1.2B [3] and MatRIS-M [4]. Experimental results show that JanusPipe delivers 1.42× to 1.67× throughput gains over 1F1B. Besides, JanusPipe does not require users to re-implement model code or manually manage pipeline schedules for each backbone.
>
> > Q4: Large model training and performance
>
> Yes. The larger UMA/eSEN models in Table 4 were used in the throughput, memory, and scalability evaluations reported in the paper.
>
> For the second part of the question, we want to clarify the paper’s scope. This paper evaluates system support for scalable conservative MLIP training, rather than end-to-end accuracy scaling across model sizes.
>
> Recent work suggests that large-scale MLIPs exhibit empirical scaling behavior with respect to model size, data, and compute [2]. JanusPipe makes these larger MLIPs trainable in distributed settings by avoiding OOM and enabling effective scaling. This is a significant step toward future scaling-law studies in MLIPs.
>
> > **Summary**
>
> We appreciate the reviewer’s valuable feedback and will incorporate the above clarifications and results into the final revision. We view JanusPipe as a distributed training system for scalable conservative MLIP training. We will open-source the code with detailed documentation. We hope this will support future research on large-scale conservative MLIPs, including scaling-law studies.
>
>
>
> [1] Filippo Bigi et al., The Dark Side of the Forces: Assessing Non-Conservative Force Models for Atomistic Machine Learning, ICML 2025.
>
> [2] Brandon M. Wood et al., UMA: A Family of Universal Models for Atoms, NeurIPS 2025.
>
> [3] Filippo Bigi et al., Pushing the Limits of Unconstrained Machine-Learned Interatomic Potentials, arXiv 2026.
>
> [4] Yuanchang Zhou et al., MatRIS: Toward Reliable and Efficient Pretrained Machine Learning Interatomic Potentials, ICLR 2026.

---

> > ### Author Rebuttal · Reviewer_MEbs · 2026-04-04
> >
> > Thank you very much for the response.
> >
> > I appreciate the additional experiments with MatRIS and PET. Regarding my point which wasn't fully answered: "This paper evaluates system support for scalable conservative MLIP training, rather than end-to-end accuracy scaling across model sizes" While I know that scalable conservative MLIP training is the goal, it would be very important that such an approach preserves accuracy (but the training is faster), and/or improves accuracy because one can train longer for the same cost. This would also be important to see in problems where conservative MLIPs may be helpful, such as running MD, computing Hessians, etc. (vs. non-conservative MLIPs).This would help strengthen the practical impact further.

---

> > > ### Author Response · Authors · 2026-04-05
> > >
> > > Thank you very much for the follow-up comments.
> > >
> > > We sincerely appreciate the reviewer’s continued focus on the practical impact of JanusPipe. We believe questions like this are particularly valuable, because they directly connect systems research to the needs of the MLIP community and to the downstream scientific uses that ultimately matter in practice.
> > >
> > > **This is exactly the motivation behind JanusPipe: to address a real scalability bottleneck that limits the practical training of MLIPs.**
> > >
> > > Recent work such as UMA reports empirical scaling behavior of MLIPs with respect to model size, data, and compute, while PET provides additional evidence that larger-scale MLIP training settings can be beneficial in practice. In this context, **JanusPipe’s value is to provide a distributed training system that partitions model across devices, thereby alleviating memory bottlenecks, avoiding out-of-memory failures, and enabling larger-scale MLIP training that would otherwise be infeasible.** We view this as an important systems step toward future large-scale studies and applications of MLIPs.
> > >
> > > > Q1: Does JanusPipe make the same model train faster without hurting accuracy? Does it improve final accuracy under the same training cost?
> > >
> > > For already small or otherwise easily trainable MLIP models, this is not the main goal of JanusPipe. Due to the overhead of distributed execution, we do not claim that JanusPipe will always reduce end-to-end training time for the same small model, nor do we claim accuracy gains at the same training cost. Instead, our goal is to support larger and more memory-demanding conservative MLIP training settings that are difficult or infeasible with single-device or straightforward data-parallel training.
> > >
> > > In our additional longer-run MatRIS experiment (see our responses to Rebuttal Q1&Q3), JanusPipe and DDP show highly consistent loss trajectories and final metrics, indicating that JanusPipe does not materially harm convergence behavior.
> > >
> > > > Q2: Does JanusPipe lead to downstream benefits for MD, Hessian computation, etc.?
> > >
> > > We agree that this is an important direction, since MLIPs are valuable in part because they can support physically meaningful downstream applications such as MD and Hessian computation. **These downstream applications therefore provide important motivation for scalable MLIP training.**
> > >
> > > Our paper focuses on the **systems side** of this problem: enabling such MLIP training to scale under realistic distributed settings. We view JanusPipe as a systems enabler that expands the feasible training regime for MLIPs.
> > >
> > > Evaluating whether larger-scale MLIP training directly improves downstream MD or Hessian-related applications would require a broader application-level study, which is beyond the scope of this paper.
> > >
> > > > The main practical impact of JanusPipe is as follows.
> > >
> > > JanusPipe’s key contribution is to make larger and more memory-intensive MLIP training settings practical by partitioning the model across devices, reducing per-device memory pressure, and enabling training regimes that would otherwise be infeasible. Beyond making such training possible, our systems design also improves training efficiency in large-scale settings while preserving training behavior.
> > >
> > > We believe this is an important systems contribution because MLIP training with four-phase double backward introduces substantial memory pressure and a unique execution pattern. JanusPipe addresses this scalability bottleneck and broadens the range of MLIP workloads that can be trained in practice.
> > >
> > > > Summary
> > >
> > > We sincerely thank the reviewer again for the thoughtful follow-up questions. We especially appreciate the reviewer’s continued focus on practical impact and downstream relevance, as this helps us clarify more precisely what JanusPipe contributes to the MLIP community.

---

### Decision · Program_Chairs · 2026-04-30

**Decision:**

Accept (regular)

**Comment:**

The submission introduces a 3D-parallel training system for conservative MLIPs, considering the specialty of the double-backward execution pattern of conservative force training, which existing distributed training pipelines do not sufficiently support. The paper introduces a set of scheduling and systems techniques, including SymFold for memory-efficient pipeline parallelism, WaveK for reducing pipeline bubbles in the four-phase execution, and GARS for handling long-tailed graph-size distributions. Across experiments on up to 32 GPUs, the method demonstrates substantial throughput and memory improvements over adapted pipeline baselines.

All reviewers agreed that the paper addresses a timely and practically relevant system problem that has received limited prior attention, and that the proposed solution is technically solid. Reviewers especially appreciated the clear formulation of the four-phase conservative MLIP workload, the elegant transformation-based design of the scheduling method, and the breadth of the empirical evaluation, including ablations, sensitivity studies, and multiple parallelism settings.

The reviewers also raised a few concerns, including stronger evidence across additional model families and hardware settings, results on larger scales, and more discussion of end-to-end impact on model quality and downstream applications. The rebuttal addressed most of these concerns, and remaining issues could be reasonable future work. In all, I believe this is a work worth the community's attention.